# CONTEXT-ENRICHED MOLECULE REPRESENTATIONS IMPROVE FEW-SHOT DRUG DISCOVERY

**Johannes Schimunek[1], Philipp Seidl[1], Lukas Friedrich[2], Daniel Kuhn[2],**
**Friedrich Rippmann[2], Sepp Hochreiter[1], and Günter Klambauer[1]**

[1] ELLIS Unit Linz and LIT AI Lab, Institute for Machine Learning, Johannes Kepler University
Linz, Austria
`schimunek@ml.jku.at`
[2] Computational Chemistry & Biologics, Merck Healthcare, Darmstadt, Germany

## ABSTRACT

A central task in computational drug discovery is to construct models from known active molecules to find further promising molecules for subsequent screening. However, typically only very few active molecules are known. Therefore, few-shot learning methods have the potential to improve the effectiveness of this critical phase of the drug discovery process. We introduce a new method for few-shot drug discovery. Its main idea is to enrich a molecule representation by knowledge about known context or reference molecules. Our novel concept for molecule representation enrichment is to associate molecules from both the support set and the query set with a large set of reference (context) molecules through a modern Hopfield network. Intuitively, this enrichment step is analogous to a human expert who would associate a given molecule with familiar molecules whose properties are known. The enrichment step reinforces and amplifies the covariance structure of the data, while simultaneously removing spurious correlations arising from the decoration of molecules. Our approach is compared with other few-shot methods for drug discovery on the FS-Mol benchmark dataset. On FS-Mol, our approach outperforms all compared methods and therefore sets a new state-of-the art for few-shot learning in drug discovery. An ablation study shows that the enrichment step of our method is the key to improve the predictive quality. In a domain shift experiment, we further demonstrate the robustness of our method. Code is available at https://github.com/ml-jku/MHNfs.

## 1 INTRODUCTION

To improve human health, combat diseases, and tackle pandemics there is a steady need of discovering new drugs in a fast and efficient way. However, the drug discovery process is time-consuming and cost-intensive (Arrowsmith, 2011). Deep learning methods have been shown to reduce time and costs of this process (Chen et al., 2018; Walters and Barzilay, 2021). They diminish the required number of both wet-lab measurements and molecules that must be synthesized (Merk et al., 2018; Schneider et al., 2020). However, as of now, deep learning approaches use only the molecular information about the ligands after being trained on a large training set. At inference time, they yield highly accurate property and activity prediction (Mayr et al., 2018; Yang et al., 2019), generative (Segler et al., 2018a; Gómez-Bombarelli et al., 2018), or synthesis models (Segler et al., 2018b; Seidl et al., 2022).

**Deep learning methods in drug discovery usually require large amounts of biological measurements.** To train deep learning-based activity and property prediction models with high predictive performance, hundreds or thousands of data points per task are required. For example, well-performing predictive models for activity prediction tasks of ChEMBL have been trained with an average of 3,621 activity points per task – i.e., drug target – by Mayr et al. (2018). The ExCAPE-DB dataset provides on average 42,501 measurements per task (Sun et al., 2017; Sturm et al., 2020). Wu et al. (2018) published a large scale benchmark for molecular machine learning, including prediction models for the SIDER dataset (Kuhn et al., 2016) with an average of 5,187 data points, Tox21 (Huang et al., 2016b; Mayr et al., 2016) with on average 9,031, and ClinTox (Wu et al., 2018) with 1,491 measurements

per task. However, for typical drug design projects, the amount of available measurements is very limited (Stanley et al., 2021; Waring et al., 2015; Hochreiter et al., 2018), since in-vitro experiments are expensive and time-consuming. Therefore, methods that need only few measurements to build precise prediction models are desirable. This problem – i.e., the challenge of learning from few data points – is the focus of machine learning areas like meta-learning (Schmidhuber, 1987; Bengio et al., 1991; Hochreiter et al., 2001) and few-shot learning (Miller et al., 2000; Bendre et al., 2020; Wang et al., 2020).

**Few-shot learning tackles the low-data problem that is ubiquitous in drug discovery.** Few-shot learning methods have been predominantly developed and tested on image datasets (Bendre et al., 2020; Wang et al., 2020) and have recently been adapted to drug discovery problems (Altae-Tran et al., 2017; Guo et al., 2021; Wang et al., 2021; Stanley et al., 2021; Chen et al., 2022). They are usually categorized into three groups according to their main approach (Bendre et al., 2020; Wang et al., 2020; Adler et al., 2020). a) Data-augmentation-based approaches augment the available samples and generate new, more diverse data points (Chen et al., 2020; Zhao et al., 2019; Antoniou and Storkey, 2019). b) Embedding-based and nearest neighbour approaches learn embedding space representations. Predictive models can then be constructed from only few data points by comparing these embeddings. For example, in Matching Networks (Vinyals et al., 2016) an attention mechanism that relies on embeddings is the basis for the predictions. Prototypical Networks (Snell et al., 2017) create prototype representations for each class using the above mentioned representations in the embedding space. c) Optimization-based or fine-tuning methods utilize a meta-optimizer that focuses on efficiently navigating the parameter space. For example, with MAML the meta-optimizer learns initial weights that can be adapted to a novel task by few optimization steps (Finn et al., 2017).

Most of these approaches have already been applied to few-shot drug discovery (see Section 4). Surprisingly, almost all these few-shot learning methods in drug discovery are worse than a naive baseline, which does not even use the support set (see Section 5). We hypothesize that the underperformance of these methods stems from disregarding the context – both in terms of similar molecules and similar activities. Therefore, we propose a method that informs the representations of the query and support set with a large number of context molecules covering the chemical space.

**Enriching molecule representations with context using associative memories.** In data-scarce situations, humans extract co-occurrences and covariances by associating current perceptions with memories (Bonner and Epstein, 2021; Potter, 2012). When we show a small set of active molecules to a human expert in drug discovery, the expert associates them with known molecules to suggest further active molecules (Gomez, 2018; He et al., 2021). In an analogous manner, our novel concept for few-shot learning uses associative memories to extract co-occurrences and the covariance structure of the original data and to amplify them in the representations (Fürst et al., 2022). We use Modern Hopfield Networks (MHNs) as an associative memory, since they can store a large set of context molecule representations (Ramsauer et al., 2021, Theorem 3). The representations that are retrieved from the MHNs replace the original representations of the query and support set molecules. Those retrieved representations have amplified co-occurrences and covariance structures, while peculiarities and spurious co-occurrences of the query and support set molecules are averaged out.

In this work, our contributions are the following:

- We propose a new architecture **MHNfs** for few-shot learning in drug discovery.
- We achieve a new state-of-the-art on the benchmarking dataset FS-Mol.
- We introduce a novel concept to enrich the molecule representations with context by associating them with a large set of context molecules.
- We add a naive baseline to the FS-Mol benchmark that yields better results than almost all other published few-shot learning methods.
- We provide results of an ablation study and a domain shift experiment to further demonstrate the effectiveness of our new method.

## 2 PROBLEM SETTING

Drug discovery projects revolve around models $g(\boldsymbol{m})$ that can predict a molecular property or activity $\hat{y}$, given a representation $\boldsymbol{m}$ of an input molecule from a chemical space $\mathcal{M}$. We consider machine learning models $\hat{y} = g_{\boldsymbol{w}}(\boldsymbol{m})$ with parameters $\boldsymbol{w}$ that have been selected using a training set. Typically,

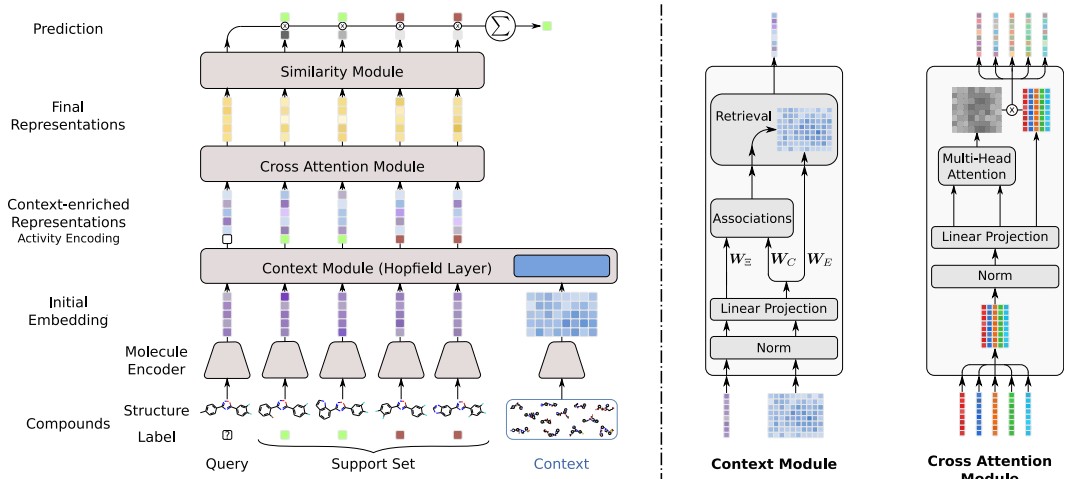

Figure 1: Schematic overview of our architecture. **Left:** All molecules are fed through a shared molecule encoder to obtain embeddings. Then, the context module (CM) enriches the representations by associating them with context molecules. The cross-attention module (CAM) enriches representations by mutually associating the query and support set molecules. Finally, the similarity module computes the prediction for the query molecule. **Right:** Detailed depiction of the operations in the CM and the CAM.

deep learning based property prediction uses a molecule encoder $f^{\mathrm{ME}} : \mathcal{M} \rightarrow \mathbb{R}^d$. The molecule encoder can process different symbolic or low-level representations of molecules, such as molecular descriptors (Bender et al., 2004; Unterthiner et al., 2014; Mayr et al., 2016), SMILES (Weininger, 1988; Mayr et al., 2018; Winter et al., 2019; Segler et al., 2018a), or molecular graphs (Merkwirth and Lengauer, 2005; Kearnes et al., 2016; Yang et al., 2019; Jiang et al., 2021) and can be pre-trained on related property prediction tasks.

For few-shot learning, the goal is to select a high-quality predictive model based on a small set of molecules $\{\boldsymbol{x}_1, \ldots, \boldsymbol{x}_N\}$ with associated measurements $\boldsymbol{y} = \{y_1, \ldots, y_N\}$. The measurements are usually assumed to be binary $y_n \in \{-1, 1\}$, corresponding to the molecule being inactive or active. The set $\{(\boldsymbol{x}_n, y_n)\}_{n=1}^N$ is called the *support set* that contains samples from a prediction task and $N$ is the *support set size*. The goal is to construct a model that correctly predicts $y$ for an $\boldsymbol{x}$ that is not in the support set – in other words, a model that generalizes well.

Standard supervised machine learning approaches typically just show limited predictive power at this task (Stanley et al., 2021) since they tend to overfit on the support set due to a small number of training samples. These approaches learn the parameters $\boldsymbol{w}$ of the model $g_{\boldsymbol{w}}$ from the support set in a supervised manner. However, they heavily overfit to the support set when $N$ is small. Therefore, few-shot learning methods are necessary to construct models from the support set that generalize well to new data.

## 3 MHNFS: HOPFIELD-BASED MOLECULAR CONTEXT ENRICHMENT FOR FEW-SHOT DRUG DISCOVERY

We aim at increasing the generalization capabilities of few-shot learning methods in drug discovery by enriching the molecule representations with molecular context. In comparison to the support set, which encodes information about the task, the context set – i.e. a large set of molecules – includes information about a large chemical space. The query and the support set molecules perform a retrieval from the context set and thereby enrich their representations. We detail this in the following.

### 3.1 MODEL ARCHITECTURE

We propose an architecture which consists of three consecutive modules. The first module – a) the *context module* $f^{\mathrm{CM}}$ – enriches molecule representations by retrieving from a large set of molecules.

The second module – b) the *cross-attention module* $f^{\mathrm{CAM}}$ (Hou et al., 2019; Chen et al., 2021) – enables the effective exchange of information between the query molecule and the support set molecules. Finally the prediction for the query molecule is computed by using the usual c) *similarity module* $f^{\mathrm{SM}}$ (Koch et al., 2015; Altae-Tran et al., 2017):

$$\text{context module:} \qquad \boldsymbol{m}' = f^{\mathrm{CM}}(\boldsymbol{m}, \boldsymbol{C})$$
$$\boldsymbol{X}' = f^{\mathrm{CM}}(\boldsymbol{X}, \boldsymbol{C}), \qquad (1)$$
$$\text{cross-attention module:} \quad [\boldsymbol{m}'', \boldsymbol{X}''] = f^{\mathrm{CAM}}([\boldsymbol{m}', \boldsymbol{X}']), \qquad (2)$$
$$\text{similarity module:} \qquad \hat{y} = f^{\mathrm{SM}}(\boldsymbol{m}'', \boldsymbol{X}'', \boldsymbol{y}), \qquad (3)$$

where $\boldsymbol{m} \in \mathbb{R}^d$ is a molecule embedding from a trainable or fixed molecule encoder, and $\boldsymbol{m}'$ and $\boldsymbol{m}''$ are enriched versions of it. Similarly, $\boldsymbol{X} \in \mathbb{R}^{d \times N}$ contains the stacked embeddings of the support set molecules and $\boldsymbol{X}'$ and $\boldsymbol{X}''$ are their enriched versions. $\boldsymbol{C} \in \mathbb{R}^{d \times M}$ is a large set of stacked molecule embeddings, $\boldsymbol{y}$ are the support set labels, and $\hat{y}$ is the prediction for the query molecule. Square brackets indicate concatenation, for example $[\boldsymbol{m}', \boldsymbol{X}']$ is a matrix with $N + 1$ columns. The modules $f^{\mathrm{CM}}$, $f^{\mathrm{CAM}}$, and $f^{\mathrm{SM}}$ are detailed in the paragraphs below. An overview of our architecture is given in Figure 1. The architecture also includes skip connections bypassing $f^{\mathrm{CM}}(.,.)$ and $f^{\mathrm{CAM}}(.)$ and layer normalization (Ba et al., 2016), which are not shown in Figure1.

A shared molecule encoder $f^{\mathrm{ME}}$ creates embeddings for the query molecule $\boldsymbol{m} = f^{\mathrm{ME}}(m)$, the support set molecules $\boldsymbol{x}_n = f^{\mathrm{ME}}(x_n)$, and the context molecules $\boldsymbol{c}_m = f^{\mathrm{ME}}(c_m)$. There are many possible choices for fixed or adaptive molecule encoders (see Section 2), of which we use descriptor-based fully-connected networks because of their computational efficiency and good accuracy (Dahl et al., 2014; Mayr et al., 2016; 2018). For notational clarity we denote the course of the representations through the architecture:

$$\underset{\substack{\text{symbolic or}\\\text{low-level repr.}}}{m} \xrightarrow{f^{\mathrm{ME}}} \underset{\substack{\text{molecule}\\\text{embedding}}}{\boldsymbol{m}} \xrightarrow{f^{\mathrm{CM}}} \underset{\substack{\text{context}\\\text{repr.}}}{\boldsymbol{m}'} \xrightarrow{f^{\mathrm{CAM}}} \underset{\substack{\text{similarity}\\\text{repr.}}}{\boldsymbol{m}''}, \qquad (4)$$

$$\underset{\substack{\text{symbolic or}\\\text{low-level repr.}}}{x_n} \xrightarrow{f^{\mathrm{ME}}} \underset{\substack{\text{molecule}\\\text{embedding}}}{\boldsymbol{x}_n} \xrightarrow{f^{\mathrm{CM}}} \underset{\substack{\text{context}\\\text{repr.}}}{\boldsymbol{x}_n'} \xrightarrow{f^{\mathrm{CAM}}} \underset{\substack{\text{similarity}\\\text{repr.}}}{\boldsymbol{x}_n''}. \qquad (5)$$

## 3.2 CONTEXT MODULE (CM)

The context module associates the query and support set molecules with a large set of context molecules, and represents them as weighted average of context molecule embeddings. The context module is realised by a continuous Modern Hopfield Network (MHN) (Ramsauer et al., 2021). An MHN is a content-addressable associative memory which can be built into deep learning architectures. There exists an analogy between the energy update of MHNs and the attention mechanism of Transformers (Vaswani et al., 2017; Ramsauer et al., 2021). MHNs are capable of storing and retrieving patterns from a memory $\boldsymbol{M} \in \mathbb{R}^{e \times M}$ given a state pattern $\boldsymbol{\xi} \in \mathbb{R}^e$ that represents the query. The retrieved pattern $\boldsymbol{\xi}^{\mathrm{new}} \in \mathbb{R}^e$ is obtained by

$$\boldsymbol{\xi}^{\mathrm{new}} = \boldsymbol{M}\,\boldsymbol{p} = \boldsymbol{M}\,\mathrm{softmax}\left(\beta \boldsymbol{M}^T \boldsymbol{\xi}\right), \qquad (6)$$

where $\boldsymbol{p}$ is called the vector of associations and $\beta$ is a scaling factor or inverse temperature. Modern Hopfield Networks have been successfully applied to chemistry and computational immunology (Seidl et al., 2022; Widrich et al., 2020).

We use this mechanism in the form of a *Hopfield layer*, which first maps raw patterns to an associative space using linear transformations, and uses multiple simultaneous queries $\boldsymbol{\Xi} \in \mathbb{R}^{d \times N}$:

$$\mathrm{Hopfield}(\boldsymbol{\Xi}, \boldsymbol{C}) := (\boldsymbol{W}_E \boldsymbol{C})\,\mathrm{softmax}\left(\beta\,(\boldsymbol{W}_C \boldsymbol{C})^T\,(\boldsymbol{W}_\Xi \boldsymbol{\Xi})\right), \qquad (7)$$

where $\boldsymbol{W}_E \in \mathbb{R}^{d \times d}$ and $\boldsymbol{W}_C, \boldsymbol{W}_\Xi \in \mathbb{R}^{e \times d}$ are trainable parameters of the Hopfield layer, softmax is applied column-wise, and $\beta$ is a hyperparameter. Note that in principle the $\boldsymbol{\Xi}$ and $\boldsymbol{C}$ could have a

different second dimension as long as the linear transformations map to the same dimension $e$. Note that all embeddings that enter this module are first layer normalized (Ba et al., 2016). Several of these Hopfield layers can run in parallel and we refer to them as "heads" in analogy to Transformers (Vaswani et al., 2017).

The context module of our new architecture uses a Hopfield layer, where the query patterns are the embeddings of the query molecule $m$ and the support set molecules $X$. The memory is composed of embeddings of a large set of $M$ molecules from a chemical space, for example reference molecules, here called context molecules $C$. Then the original embeddings $m$ and $X$ are replaced by the retrieved embeddings, which are weighted averages of context molecule embeddings:

$$m' = \text{Hopfield}(m, C) \quad \text{and} \quad X' = \text{Hopfield}(X, C). \tag{8}$$

This retrieval step reinforces the covariance structure of the retrieved representations (see Appendix A.8), which usually enhances robustness of the models (Fürst et al., 2022) by removing noise. Note that the embeddings of the query and the support set molecules have not yet influenced each other. These updated representations $m'$, $X'$ are passed to the cross-attention module. Exemplary retrievals from the context module are included in Appendix A.7.

## 3.3 CROSS-ATTENTION MODULE (CAM)

For embedding-based few-shot learning methods in the field of drug discovery, Altae-Tran et al. (2017) showed that the representations of the molecules can be enriched, if the architecture allows information exchange between query and support set molecules. Altae-Tran et al. (2017) uses an attention-enhanced LSTM variant, which updates the query and the support set molecule representations in an iterative fashion being aware of each other. We further develop this idea and combine it with the idea of using a transformer encoder layer (Vaswani et al., 2017) as a cross-attention module (Hou et al., 2019; Chen et al., 2021).

The cross-attention module updates the query molecule representation $m'$ and the support set molecule representations $X'$ by mutually exchanging information, using the usual Transformer mechanism:

$$[m'', X''] = \text{Hopfield}([m', X'], [m', X']), \tag{9}$$

where $[m', X'] \in \mathbb{R}^{d \times (N+1)}$ is the concatenation of the representations of the query molecule $m'$ with the support set molecules $X'$ and we exploited that the Transformer is a special case of the Hopfield layer. Again, normalization is applied (Ba et al., 2016) and multiple Hopfield layers – i.e., heads – can run in parallel, be stacked, and equipped with skip-connections. The representations $m''$ and $X''$ are passed to the similarity module.

## 3.4 SIMILARITY MODULE (SM)

In this module, pairwise similarity values $k(m'', x_n'')$ are computed between the representation of a query molecule $m''$ and each molecule $x_n''$ in the support set as done recently (Koch et al., 2015; Altae-Tran et al., 2017). Based on these similarity values, the activity for the query molecule is predicted, building a weighted mean over the support set labels:

$$\hat{y} = \sigma \left( \tau^{-1} \frac{1}{N} \sum_{n=1}^{N} y_n' \, k(m'', x_n'') \right), \tag{10}$$

where our architecture employs dot product similarity of normalized representations $k(m'', x_n'') = m''^T x_n''$. $\sigma(.)$ is the sigmoid function and $\tau$ is a hyperparameter. Note that we use a balancing strategy for the labels $y_n' = \begin{cases} N/(\sqrt{2N_A}) & \text{if } y_n = 1 \\ -N/(\sqrt{2N_I}) & \text{else} \end{cases}$, where $N_A$ is the number of actives and $N_I$ is the number of inactives of the support set.

## 3.5 ARCHITECTURE, HYPERPARAMETER SELECTION, AND TRAINING DETAILS

**Hyperparameters.** The main hyperparameters of our architecture are the number of heads, the embedding dimension, the dimension of the association space of the CAM and CM, the learning

rate schedule, the scaling parameter $\beta$, and the molecule encoder. The following hyperparameters were selected by manual hyperparameter selection on the validation tasks. The molecule encoder consists of a single layer with output size $d = 1024$ and SELU activation (Klambauer et al., 2017). The CM consists of one Hopfield layer with 8 heads. The dimension $e$ of the association space is set to 512 and $\beta = 1/\sqrt{e}$. Since we use skip connections between all modules the output dimension of the CM and CAM matches the input dimension. The CAM comprises one layer with 8 heads and an association-space dimension of 1088. For the input to the CAM, an activity encoding was added to the support set molecule representations to provide label information. The SM uses $\tau = 32$. For the context set, we randomly sample 5% from a large set of molecules – i.e., the molecules in the FS-Mol training split – for each batch. For inference, we used a fixed set of 5% of training set molecules as the context set for each seed. We hypothesize that these choices about the context could be further improved (Section 6). We provide considered and selected hyperparameters in Appendix A.1.6.

**Loss function, regularization and optimization.** We use the Adam optimizer (Kingma and Ba, 2014) to minimize the cross-entropy loss between the predicted and known activity labels. We use a learning rate scheduler which includes a warm up phase, followed by a section with a constant learning rate, which is 0.0001, and a third phase in which the learning rate steadily decreases. As a regularization strategy, for the CM and the CAM a dropout rate of 0.5 is used. The molecule encoder has a dropout with rate 0.1 for the input and 0.5 elsewhere (see also Appendix A.1.6).

**Compute time and resources.** Training a single **MHNfs** model on the benchmarking dataset FS-Mol takes roughly 90 hours of wall-clock time on an A100 GPU. In total, roughly 15,000 GPU hours were consumed for this work.

## 4 RELATED WORK

Several approaches to few-shot learning in drug discovery have been suggested (Altae-Tran et al., 2017; Nguyen et al., 2020; Guo et al., 2021; Wang et al., 2021). Nguyen et al. (2020) evaluated the applicability of MAML and its variants to graph neural networks (GNNs) and (Guo et al., 2021) also combine GNNs and meta-learning. Altae-Tran et al. (2017) suggested an approach called Iterative Refinement Long Short-Term Memory, in which query and support set embeddings can share information and update their embeddings. Property-aware relation networks (PAR) (Wang et al., 2021) use an attention mechanism to enrich representations from cluster centers and then learn a relation graph between molecules. Chen et al. (2022) propose to adaptively learn kernels and apply their method to few-shot drug discovery with predictive performance for larger support set sizes. Recently, Stanley et al. (2021) generated a benchmark dataset for few-shot learning methods in drug discovery and provided some baseline results.

Many successful deep neural network architectures use external memories, such as the neural Turing machine (Graves et al., 2014), memory networks (Weston et al., 2014), end-to-end memory networks (Sukhbaatar et al., 2015). Recently, the connection between continuous modern Hopfield networks (Ramsauer et al., 2021), which are content-addressable associative memories, and Transformer architectures (Vaswani et al., 2017) has been established. We refer to Le (2021) for an extensive overview of memory-based architectures. Architectures with external memories have also been used for meta-learning (Vinyals et al., 2016; Santoro et al., 2016) and few-shot learning (Munkhdalai and Yu, 2017; Ramalho and Garnelo, 2018; Ma et al., 2021).

## 5 EXPERIMENTS

### 5.1 BENCHMARKING ON FS-MOL

**Experimental setup.** Recently, the dataset FS-Mol (Stanley et al., 2021) was proposed to benchmark few-shot learning methods in drug discovery. It was extracted from ChEMBL27 and comprises in total 489,133 measurements, 233,786 compounds and 5,120 tasks. Per task, the mean number of data points is 94. The dataset is well balanced as the mean ratio of active and inactive molecules is close to 1. The FS-Mol benchmark dataset defines 4,938 training, 40 validation and 157 test tasks, guaranteeing disjoint task sets. Stanley et al. (2021) precomputed extended connectivity fingerprints (ECFP) (Rogers and Hahn, 2010) and key molecular physical descriptors, which were defined by RDKit (Landrum et al., 2006). While methods would be allowed to use other representations of

Table 1: Results on FS-MOL [ΔAUC-PR]. The best method is marked bold. Error bars represent standard errors across tasks according to Stanley et al. (2021). The metrics are also averaged across five training reruns and ten draws of support sets. In brackets the number of tasks per category is reported.

| Method | All [157] | Kin. [125] | Hydrol. [20] | Oxid.[7] |
|---|---|---|---|---|
| GNN-ST[a] (Stanley et al., 2021) | $.029 \pm .004$ | $.027 \pm .004$ | $.040 \pm .018$ | $.020 \pm .016$ |
| MAT[a] (Maziarka et al., 2020) | $.052 \pm .005$ | $.043 \pm .005$ | $.095 \pm .019$ | $.062 \pm .024$ |
| Random Forest[a] (Breiman, 2001) | $.092 \pm .007$ | $.081 \pm .009$ | $.158 \pm .028$ | $.080 \pm .029$ |
| GNN-MT[a] (Stanley et al., 2021) | $.093 \pm .006$ | $.093 \pm .006$ | $.108 \pm .025$ | $.053 \pm .018$ |
| Similarity Search | $.118 \pm .008$ | $.109 \pm .008$ | $.166 \pm .029$ | $.097 \pm .033$ |
| GNN-MAML[a] (Stanley et al., 2021) | $.159 \pm .009$ | $.177 \pm .009$ | $.105 \pm .024$ | $.054 \pm .028$ |
| PAR (Wang et al., 2021) | $.164 \pm .008$ | $.182 \pm .009$ | $.109 \pm .020$ | $.039 \pm .008$ |
| Frequent Hitters | $.182 \pm .010$ | $.207 \pm .009$ | $.098 \pm .009$ | $.041 \pm .005$ |
| ProtoNet[a] (Snell et al., 2017) | $.207 \pm .008$ | $.215 \pm .009$ | $.209 \pm .030$ | $.095 \pm .029$ |
| Siamese Networks (Koch et al., 2015) | $.223 \pm .010$ | $.241 \pm .010$ | $.178 \pm .026$ | $.082 \pm .025$ |
| IterRefLSTM (Altae-Tran et al., 2017) | $.234 \pm .010$ | $.251 \pm .010$ | $.199 \pm .026$ | $.098 \pm .027$ |
| ADKF-IFT[b] (Chen et al., 2022) | $.234 \pm .009$ | $.248 \pm .020$ | $\mathbf{.217} \pm .017$ | $\mathbf{.106} \pm .008$ |
| MHNfs (ours) | $\mathbf{.241} \pm .009$ | $\mathbf{.259} \pm .010$ | $.199 \pm .027$ | $.096 \pm .019$ |

[a] metrics from Stanley et al. (2021).    [b] results from Chen et al. (2022).

the input molecules, such as the molecular graph, we used a concatenation of these ECFPs and RDKit-based descriptors. We use the main benchmark setting of FS-Mol with support set size 16, which is close to the 5- and 10-shot settings in computer vision, and stratified random split (Stanley et al., 2021, Table 2) for a fair method comparison (see also Section A.5).

**Methods compared.** Baselines for few-shot learning and our proposed method **MHNfs** were compared against each other. The **Frequent Hitters** model is a naive baseline that ignores the provided support set and therefore has to learn to predict the average activity of a molecule. This method can potentially discriminate so-called frequent-hitter molecules (Stork et al., 2019) against molecules that are inactive across many tasks. We also added **Similarity Search** (Cereto-Massagué et al., 2015) as a baseline. Similarity search is a standard chemoinformatics technique, used in situations with single or few known actives. In the simplest case, the search finds similar molecules by computing a fingerprint or descriptor-representation of the molecules and using a similarity measure $k(.,.)$ – such as Tanimoto Similarity (Tanimoto, 1960). Thus, Similarity Search, as used in chemoinformatics, can be formally written as $\hat{y} = 1/N \sum_{n=1}^{N} y_n \, k(\boldsymbol{m}, \boldsymbol{x}_n)$, where $\boldsymbol{x}_1, \ldots, \boldsymbol{x}_n$ come from a fixed molecule encoder, such as chemical fingerprint or descriptor calculation. A natural extension of Similarity Search with fixed chemical descriptors is **Neural Similarity Search or Siamese networks** (Koch et al., 2015), which extend the classic similarity search by learning a molecule encoder: $\hat{y} = \sigma\left(\tau^{-1} \frac{1}{N} \sum_{n=1}^{N} y'_n \, f_{\boldsymbol{w}}^{\text{ME}}(\boldsymbol{m})^T \, f_{\boldsymbol{w}}^{\text{ME}}(\boldsymbol{x}_n)\right)$. Furthermore, we re-implemented the **IterRefLSTM** (Altae-Tran et al., 2017) in PyTorch. The **IterRefLSTM** model consists of three modules. First, a molecule encoder maps the query and support set molecules to its representations $\boldsymbol{m}$ and $\boldsymbol{X}$. Second, an attention-enhanced LSTM variant, the actual **IterRefLSTM**, iteratively updates the query and support set molecules, enabling information sharing between the molecules: $[\boldsymbol{m}', \boldsymbol{X}'] = \text{IterRefLSTM}_L([\boldsymbol{m}, \boldsymbol{X}])$, where the hyperparameter $L$ controls the number of iteration steps of the **IterRefLSTM**. Third, a similarity module computes attention weights based on the representations: $\boldsymbol{a} = \text{softmax}(k(\boldsymbol{m}', \boldsymbol{X}'))$. These representations are then used for the final prediction: $\hat{y} = \sum_{i=1}^{N} a_i y_i$. For further details, see Appendix A.1.5. The **Random Forest** baseline uses the chemical descriptors and is trained in standard supervised manner on the support set molecules for each task. The method **GNN-ST** is a graph neural network (Stanley et al., 2021; Gilmer et al., 2017) that is trained from scratch for each task. The **GNN-MT** uses a two step strategy: First, the model is pretrained on a large dataset on related tasks; second, an output layer is constructed to the few-shot task via linear probing (Stanley et al., 2021; Alain and Bengio, 2016). The **Molecule Attention Transformer (MAT)** is pre-trained in a self-supervised fashion and fine-tuning is performed for the few-shot task (Maziarka et al., 2020). **GNN-MAML** is based on MAML (Finn et al., 2017), and uses a model-agnostic meta-learning strategy to find a general core model from which one can easily adapt to single tasks. Notably, GNN-MAML also can be seen as a proxy for **Meta-MGNN** (Guo

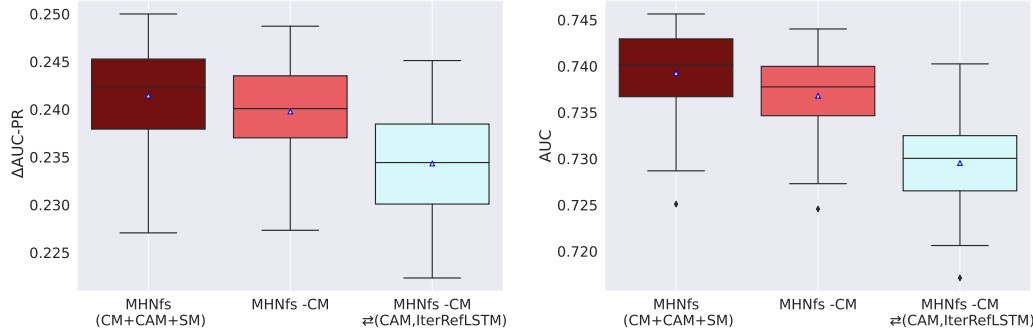

Figure 2: Results of the ablation study. The boxes show the median, mean and the variability of the average predictive performance of the methods across training reruns and draws of support sets. The performance significantly drops when the context module is removed (light red bars), and when additionally the cross-attention module is replaced with the **IterRefLSTM** module (light blue bars). This indicates that our two newly introduced modules, CM and CAM, play a crucial role in MHNfs.

et al., 2021), which enriches the gradient update step in the outer loop of the MAML-framework by an attention mechanism and uses an additional atom type prediction loss and a bond reconstruction loss. **ProtoNet** (Snell et al., 2017) includes a molecule encoder, which maps query and support set molecules to representations in an embedding space. In this embedding space, prototypical representations of each class are built by taking the mean across all related support set molecules for each class (details in Appendix A.1.4). The **PAR** model (Wang et al., 2021) includes a GNN which creates initial molecule embeddings. These molecule embeddings are then enriched by an attention mechanism. Finally, another GNN learns relations between support and query set molecules. The **PAR** model has shown good results for datasets which just include very few tasks such as Tox21 (Wang et al., 2021). Chen et al. (2022) suggest a framework for learning deep kernels by interpolating between meta-learning and conventional deep kernels, which results in the **ADKF-IFT** model. The model has exhibited especially high performance for large support set sizes. For all methods the most important hyperparameters were adjusted on the validation tasks of FS-Mol.

**Training and evaluation.** For the model implementations, we used PyTorch (Paszke et al., 2019, BSD license). We used PyTorch Lightning (Falcon et al., 2019, Apache 2.0 license) as a framework for training and test logic, hydra for config file handling (Yadan, 2019, Apache 2.0 license) and Weights & Biases (Biewald, 2020, MIT license) as an experiment tracking tool. We performed five training reruns with different seeds for all methods, except Classic Similarity Search as there is no variability across seeds. Each model was evaluated ten times by drawing support sets with ten different seeds.

**Results.** The results in terms of area under precision-recall curve (AUC-PR) are presented in Table 1, where the difference to a random classifier is reported ($\Delta$AUC-PR). The standard error is reported across tasks. Surprisingly, the naive baseline **Frequent Hitters**, that neglects the support set, has outperformed most of the few-shot learning methods, except for the embedding based methods and ADKF-IFT. **MHNfs** has outperformed all other methods with respect to $\Delta$AUC-PR across all tasks, including the **IterRefLSTM** model ($p$-value 1.72e-7, paired Wilcoxon test), the **ADKF-IFT** model ($p$-value <1.0e-8, Wilcoxon test), and the **PAR** model ($p$-value <1.0e-8, paired Wilcoxon test).

## 5.2 ABLATION STUDY

**MHNfs** has two new main components compared to the most similar previous state-of-the-art method **IterRefLSTM**: i) the context module, and ii) the cross-attention module which replaces the LSTM-like module. To assess the effects of these components, we performed an ablation study. Therefore, we compared **MHNfs** to a method that does not have the context module ("MHNfs -CM") and to a method that does not have the context module and uses an LSTM-like module instead of the CAM ("MHNfs -CM $\rightleftharpoons$(CAM,IterRefLSTM)"). For the ablation study, we used all 5 training reruns and evaluated 10 times on the test set with different support sets. The results of this ablation steps are presented in Figure 2. Both removing the CM and exchanging the CAM with the **IterRefLSTM** module were

Table 2: Results of the domain shift experiment on Tox21 [AUC, $\Delta$AUC-PR]. The best method is marked bold. Error bars represent standard deviation across training reruns and draws of support sets

| Method | AUC | $\Delta$AUC-PR |
|--------|-----|----------------|
| Similarity Search (baseline) | $.629 \pm .015$ | $.061 \pm .008$ |
| IterRefLSTM (Altae-Tran et al., 2017) | $.664 \pm .018$ | $.067 \pm .008$ |
| MHNfs (ours) | $\mathbf{.679} \pm .018$ | $\mathbf{.073} \pm .008$ |

detrimental for the performance of the method ($p$-value 0.002 and $1.72e-7$, respectively; paired Wilcoxon test). The difference was even more pronounced under domain shift (see Appendix A.3.3). Appendix A.3.2 contains a second ablation study that examines the overall effects of the context, the cross-attention, the similarity module, and the molecule encoder of **MHNfs**.

## 5.3 DOMAIN SHIFT EXPERIMENT

**Experimental setup.** The Tox21 dataset consists of 12,707 chemical compounds, for which measurements for up to 12 different toxic effects are reported (Mayr et al., 2016; Huang et al., 2016a). It was published with a fixed training, validation and test split. State-of-the-art supervised learning methods that have access to the full training set reach AUC performance values between $0.845$ and $0.871$ (Klambauer et al., 2017; Duvenaud et al., 2015; Li et al., 2017; 2021; Zaslavskiy et al., 2019; Alperstein et al., 2019). For our evaluation, we re-cast Tox21 as a few-shot learning setting and draw small support sets from the 12 tasks. The compared methods were pre-trained on FS-Mol and obtain small support sets from Tox21. Based on the support sets, the methods had to predict the activities of the Tox21 test set. Note that there is a strong domain shift from drug-like molecules of FS-Mol to environmental chemicals, pesticides, food additives of Tox21. The domain shift also concerns the outputs where a shift from kinases, hydrolases, and oxidoreductases of FS-Mol to nuclear receptors and stress responses of Tox21 is present.

**Methods compared.** We compared **MHNfs**, the runner-up method **IterRefLSTM**, and **Similarity Search** – since it has been widely used for such purposes for decades (Cereto-Massagué et al., 2015).

**Training and evaluation.** We followed the procedure of Stanley et al. (2021) for data-cleaning, preprocessing and extraction of the fingerprints and descriptors used in FS-Mol. After running the cleanup step, 8,423 molecules remained for the domain shift experiments. From the training set, 8 active and 8 inactive molecules per task were randomly selected to build the support set. The test set molecules were used as query molecules. The validation set molecules were not used at all. During test-time, a support set was drawn ten times for each task. Then, the performance of the models were evaluated for these support sets, using the area under precision-recall curve (AUC-PR), analogously to the FS-Mol benchmarking experiment reported as the difference to a random classifier ($\Delta$AUC-PR), and the area under receiver operating characteristic curve (AUC) metrics. The performance values report the mean over all combinations regarding the training reruns and the support set sampling iterations. Error bars indicate the standard deviation.

**Results.** The Hopfield-based context retrieval method has significantly outperformed the IterRefLSTM-based model ($p_{\Delta AUC-PR}$-value $3.4e-5$, $p_{AUC}$-value $2.5e-6$, paired Wilcoxon test) and the Classic Similarity Search ($p_{\Delta AUC-PR}$-value $2.4e-9$, $p_{AUC}$-value $7.6e-10$, paired Wilcoxon test) and therefore showed robust performance on the toxicity domain, see Table 2. Notably, all models were trained on the FS-Mol dataset and then applied to the Tox21 dataset without adjusting any weight parameter.

## 6 CONCLUSION

We have introduced a new architecture for few-shot learning in drug discovery, namely MHNfs, that is based on the novel concept to enrich molecule representations with context. In a benchmarking experiment the architecture outperformed all other methods and in a domain shift study the robustness and transferability has been assessed. We envision that the context module can be applied to many different areas, enriching learned representations analogously to our work. For discussion, see A.9.

ACKNOWLEDGEMENTS

The ELLIS Unit Linz, the LIT AI Lab, the Institute for Machine Learning, are supported by the Federal State Upper Austria. IARAI is supported by Here Technologies. We thank Merck Healthcare KGaA for the collaboration. Further, we thank the projects AI-MOTION (LIT-2018-6-YOU-212), DeepFlood (LIT-2019-8-YOU-213), Medical Cognitive Computing Center (MC3), INCONTROL-RL (FFG-881064), PRIMAL (FFG-873979), S3AI (FFG-872172), DL for GranularFlow (FFG-871302), EPILEPSIA (FFG-892171), AIRI FG 9-N (FWF-36284, FWF-36235), ELISE (H2020-ICT-2019-3 ID: 951847), Stars4Waters (HORIZON-CL6-2021-CLIMATE-01-01). We thank Audi.JKU Deep Learning Center, TGW LOGISTICS GROUP GMBH, Silicon Austria Labs (SAL), FILL Gesellschaft mbH, Anyline GmbH, Google, ZF Friedrichshafen AG, Robert Bosch GmbH, UCB Biopharma SRL, Verbund AG, GLS (Univ. Waterloo) Software Competence Center Hagenberg GmbH, TÜV Austria, Frauscher Sensonic and the NVIDIA Corporation.

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

## A  APPENDIX

**Contents of the appendix**

### A.1  DETAILS ON METHODS

Few-shot learning methods in drug discovery can be described as models with adaptive parameters $\boldsymbol{w}$ that use a support set $\boldsymbol{Z} = \{(\boldsymbol{x}_1, y_1), \ldots, (\boldsymbol{x}_N, y_N)\}$ [1] as additional input to predict a label $\hat{y}$ for a molecule $\boldsymbol{m}$

$$\hat{y} = g_{\boldsymbol{w}}(\boldsymbol{m}, \boldsymbol{Z}). \tag{A1}$$

Optimization-based methods, such as MAML (Finn et al., 2017), use the support set to update the parameters $\boldsymbol{w}$

$$\hat{y} = g_{a(\boldsymbol{w};\boldsymbol{Z})}(\boldsymbol{m}), \tag{A2}$$

where $a(.)$ is a function that adapts $\boldsymbol{w}$ of $g$ based on $\boldsymbol{Z}$ for example via gradient-descent.

Embedding-based methods use a different approach and learn representations of the support set molecules $\{\boldsymbol{x}_1, \ldots, \boldsymbol{x}_N\}$, sometimes written as stacked embeddings $\boldsymbol{X} \in \mathbb{R}^{d \times N}$, and the query molecule $\boldsymbol{m}$, and some function that associates these two types of information with each other. We describe the embedding-based methods Similarity Search in Section A.1.2, Neural Similarity Search in Section A.1.3, ProtoNet in Section A.1.4, IterRefLSTM in Section A.1.5, PAR in Section A.1.7, and MHNfs in the main paper and details in Section A.1.6. The "frequent hitters" baseline is described in Section A.1.1.

### A.1.1  FREQUENT HITTERS: DETAILS AND HYPERPARAMETERS

The "frequent hitters" model $g^{\mathrm{FH}}$ is a baseline that we implemented and included in the method comparison. This method uses the usual training scheme of sampling a query molecule $\boldsymbol{m}$ with a label $y$, having access to a support set $\boldsymbol{Z}$. In contrast to the usual models of the type $g_{\boldsymbol{w}}(\boldsymbol{m}, \boldsymbol{Z})$, the frequent hitters model $g^{\mathrm{FH}}$ neglects the support set:

$$\hat{y} = g_{\boldsymbol{w}}^{\mathrm{FH}}(\boldsymbol{m}). \tag{A3}$$

---

[1]We use $\boldsymbol{Z}$ to denote the support set of already embedded molecules to keep the notation uncluttered. More correctly, the methods have access to the raw support set $Z = \{(x_1, y_1), \ldots, (x_N, y_N)\}$, where $x_n$ is a symbolic, such as the molecular graph, or low-level representation of the molecule.

Table A1: Hyperparameter space considered for the Frequent Hitters model. The hyperparameters of the best configuration are marked bold.

| Hyperparameter | Explored values |
| --- | --- |
| Number of hidden layers | 1, **2**, 4 |
| Number of units per hidden layer | 1024, **2048**, 4096 |
| Output dimension | **512**, 1024 |
| Activation function | **ReLU** |
| Learning rate | **0.0001**, 0.001 |
| Optimizer | Adam,**AdamW** |
| Weight decay | 0, **0.01** |
| Batch size | 32, 128, 512, 2048, **4096** |
| Input Dropout | 0, **0.1** |
| Dropout | 0.1, 0.2, 0.3, **0.4**, 0.5 |
| Layer-normalization | False, **True** |
| • Affine | **False**, True |
| Similarity function | **dot product** |

Thus, during training for the same molecule $m$, the model might have to predict both $y = 1$ and $y = -1$, since the molecule can be active in one task and inactive in another task. Therefore, the model tends to predict average activity of a molecule to minimize the cross-entropy loss. We chose an additive combination of the Morgan fingerprints, RDKit fingerprints, and MACCS keys for the input representation to the MLP.

**Hyperparameter search.** We performed manual hyperparameter search on the validation set and report the explored hyperparameter space (Table A1). We use early-stopping based on validation average-precision, a patience of 3 epochs and train for a maximum of 20 epochs with a linear warm-up learning-rate schedule for the first 3 epochs.

A.1.2 CLASSIC SIMILARITY SEARCH: DETAILS AND HYPERPARAMETERS

Similarity Search (Cereto-Massagué et al., 2015) is a classic chemoinformatics technique used in situations in which a single or few actives are known. In the simplest case, molecules that are similar to a given active molecule are searched by computing a fingerprint or descriptor-representation $f^{\text{desc}}(m)$ of the molecules and using a similarity measure $k(.,.)$, such as Tanimoto Similarity(Tanimoto, 1960). Thus, the Similarity Search as used in chemoinformatics can be formally written as:

$$\hat{y} = 1/N \sum_{n=1}^{N} y_n \, k(f^{\text{desc}}(m), f^{\text{desc}}(x_n)), \tag{A4}$$

where the function $f^{\text{desc}}$ maps the molecule to its chemical descriptors or fingerprints and takes the role of both the molecule encoder and the support set encoder. Then, an association function, consisting of a) the similarity measure $k(.,.)$ and b) mean pooling across molecules weighted by their similarity and activity, is used to compute the predictions.

Notably, there are many variants of Similarity Search (Cereto-Massagué et al., 2015; Wang et al., 2010; Eckert and Bajorath, 2007; Geppert et al., 2008; Willett, 2014; Sheridan and Kearsley, 2002; Riniker and Landrum, 2013) of which some correspond to recent few-shot learning methods with a fixed molecule encoder. For example, (Geppert et al., 2008) suggest to use centroid molecules, i.e., prototypes or averages of active molecules. This is equivalent to the idea of Prototypical Networks (Snell et al., 2017). Riniker and Landrum (2013) are aware of different fusion strategies for sets of active or inactive molecules, which corresponds to different pooling strategies of the support set. Overall, the variants of the classic Similarity Search are highly similar to embedding-based few-shot learning methods except that they have a fixed instead of a learned molecule encoder.

**Hyperparameter search.** For the Similarity Search, there were two decisions to make which was firstly the similarity metric and secondly the question whether we should use a balancing strategy

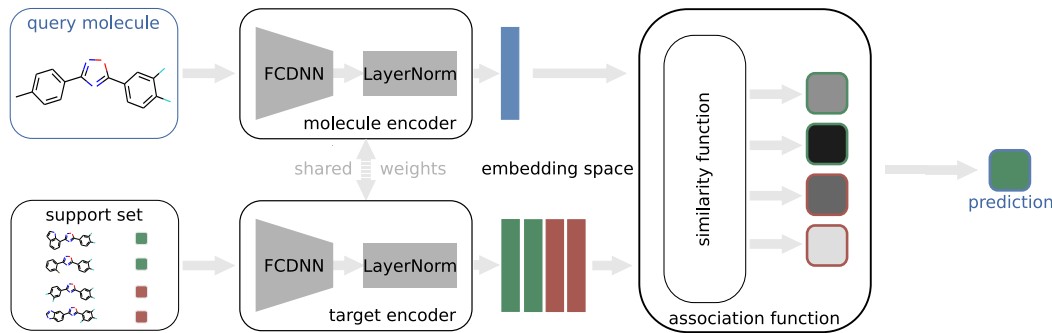

Figure A1: Schematic overview of the implemented Neural Similarity Search variant

like shown in Section 3.4. We decided for the dot-product as a similarity metric and for using the balancing strategy. These decisions were made by evaluating the models on the validation set.

### A.1.3 NEURAL SIMILARITY SEARCH OR SIAMESE NETWORKS: DETAILS AND HYPERPARAMETERS

If the fixed encoder $f^{\text{desc}}$ of the Classic Similarity Search is replaced by learned encoders $f_{\boldsymbol{w}}^{\text{ME}}$, Neural Similarity Search variants naturally arise.

A lot of related work already was done (Koch et al., 2015; Hertz et al., 2006; Ye and Guo, 2018; Torres et al., 2020). We adapted these ideas, such that a fully-connected deep neural network followed by a Layer Normalization (Ba et al., 2016) operation, $f_{\boldsymbol{w}}^{\text{ME}}$ with adaptive parameters $\boldsymbol{w}$, is used in a Siamese fashion to compute the embeddings for the input molecule and the support set molecules. Within an association function block, pairwise similarity values for the query molecule and each support set molecule are computed, associating both embeddings via the dot product. Based on these similarity values, the activity for the query molecule is predicted, building the weighted mean over the support set molecule labels:

$$\hat{y} = \sigma \left( \tau^{-1} \frac{1}{N} \sum_{n=1}^{N} y'_n \, f^{\text{ME}}(m)^T \, f^{\text{ME}}(x_n) \right), \tag{A5}$$

where $\sigma(.)$ is the sigmoid function and $\tau$ is a hyperparameter in the range of $\sqrt{d}$. Note that this method uses a balancing strategy for the labels $y'_n = \begin{cases} N/(\sqrt{2N_A}) & \text{if } y_n = 1 \\ -N/(\sqrt{2N_I}) & \text{else} \end{cases}$, where $N_A$ is the number of actives and $N_I$ is the number of inactives of the support set. Figure A1 provides an schematic overview of the Neural Similarity Search variant.

We trained the model using the Adam optimizer (Kingma and Ba, 2014) to minimize binary cross-entropy loss.

**Hyperparameter search.** We performed manual hyperparameter search on the validation set. We report the explored hyperparameter space (Table A2). Bold values indicate the selected hyperparameters for the final model.

### A.1.4 PROTONET: DETAILS AND HYPERPARAMETERS

Prototypical Networks (ProtoNet) (Snell et al., 2017) learn a prototype $\boldsymbol{r}$ for each class. Concretely, the support set $Z$ is class-wise separated into $Z^+ := \{(x, y) \in Z \mid y = 1\}$ and $Z^- := \{(x, y) \in Z \mid y = -1\}$. For the subsets $Z^+$ and $Z^-$ prototypical representations $\boldsymbol{r}^+$ and $\boldsymbol{r}^-$ can be computed by

$$\boldsymbol{r}^+ = \frac{1}{|Z^+|} \cdot \sum_{(x,y) \in Z^+} f^{\text{ME}}(x) \tag{A6}$$

Table A2: Hyperparameter space considered for the Neural Similarity Search model selection. The hyperparameters of the best configuration are marked bold.

| Hyperparameter | Explored values |
|---|---|
| Number of hidden layers | 1, **2**, 4 |
| Number of units per hidden layer | **1024**, 4096 |
| Output dimension | **512**, 1024 |
| Activation function | ReLU, **SELU** |
| Learning rate | 0.0001, **0.001**, 0.01 |
| Optimizer | **Adam** |
| Weight decay | **0**, $1 \cdot 10^{-4}$ |
| Batch size | **4096** |
| Input Dropout | **0.1** |
| Dropout | **0.5** |
| Layer-normalization | False, **True** |
| • Affine | **False** |
| Similarity function | cosine similarity, **dot product**, MinMax similarity |

and

$$\boldsymbol{r}^- = \frac{1}{|Z^-|} \cdot \sum_{(x,y) \in Z^-} f^{\mathrm{ME}}(x). \tag{A7}$$

The prototypical representations $\boldsymbol{r}^+, \boldsymbol{r}^- \in \mathbb{R}^d$ and the query molecule embedding $\boldsymbol{m} \in \mathbb{R}^d$ are then used to make the final prediction:

$$\hat{y} = \frac{\exp(-\boldsymbol{d}(\boldsymbol{m}, \boldsymbol{r}^+))}{\exp(-\boldsymbol{d}(\boldsymbol{m}, \boldsymbol{r}^+)) + \exp(-\boldsymbol{d}(\boldsymbol{m}, \boldsymbol{r}^-))}, \tag{A8}$$

where $\boldsymbol{d}$ is a distance metric.

**Hyperparameter search.** Hyperparameter search has been done by Stanley et al. (2021), to which we refer here. ECFP fingerprints and descriptors created by a GNN, which operates on the molecular graph, are fed into a fully connected neural network, which maps the input into an embedding space with the dimension of 512. Stanley et al. (2021) use the Mahalanobis distance to measure the similarity between a query molecule and the prototypical representations in the embedding space. The learning rate is 0.001 and the batch size is 256. The implementation can be found here https://github.com/microsoft/FS-Mol/blob/main/fs_mol/protonet_train.py and important hyperparameters are chosen here https://github.com/microsoft/FS-Mol/blob/main/fs_mol/utils/protonet_utils.py.

**Connection to Siamese networks and contrastive learning with InfoNCE.** If instead of the negative distance $-\boldsymbol{d}(.,.)$ the dot product similarity measure with appropriate scaling is used, ProtoNet for two classes becomes equivalent to Siamese Networks. Note that in our study, another difference is that ProtoNet uses a GNN for the encoder, whereas the encoder of the Siamese Networks is a descriptor-based fully-connected network. In case of dot product as similarity measure, the objective also becomes equivalent to contrastive learning with the InfoNCE objective (Oord et al., 2018).

### A.1.5 ITERREFLSTM: DETAILS AND HYPERPARAMETERS

Altae-Tran et al. (2017) modified the idea of Matching Networks (Vinyals et al., 2016) by replacing the LSTM with their Iterative Refinement Long Short-Term Memory (IterRefLSTM). The use of the IterRefLSTM empowers the architecture to update not only the embeddings for the query molecule but also adjust the representations of the support set molecules.

For the IterRefLSTM model, query molecule embedding $\boldsymbol{m} = f_{\boldsymbol{\theta}_1}^{\mathrm{ME}}(m)$ and support set molecule embeddings $\boldsymbol{x}_n = f_{\boldsymbol{\theta}_2}^{\mathrm{ME}}(x_n)$ are created using two potentially different molecule encoders for the query molecule $m$ and the support set molecules $x_1, \ldots, x_N$. The query and support set molecule

Table A3: Hyperparameter space considered for the IterRefLSTM model selection. The hyperparameters of the best configuration are marked bold.

| Hyperparameter | Explored values |
|---|---|
| Molecule encoder | |
| • Number of hidden layers | **0**, 1, 2, 4 |
| • Number of units per hidden layer | **1024**, 4096 |
| • Output dimension | **512**, 1024 |
| • Activation function | ReLU, **SELU** |
| • Input dropout | **0.1** |
| • Dropout | **0.5** |
| IterRef embedding layer | |
| • L | 1, **3** |
| Similarity module: | |
| • Metric | cosine similarity, **dot product**, MinMax similarity |
| • Similarity space dimension | 512, **1024** |
| Layer-normalization | False, **True** |
| • Affine | **False**, True |
| Training | |
| • Learning rate | 0.0001, **0.001**, 0.01 |
| • Optimizer | **Adam**, AdamW |
| • Weight decay | **0**, 0.0001 |
| • Batch size | **2048**, 4096 |

embeddings are then updated by an LSTM-like module – the actual IterRefLSTM:

$$[\boldsymbol{m}', \boldsymbol{X}'] = \text{IterRefLSTM}_L([\boldsymbol{m}, \boldsymbol{X}]).$$

Here, $\boldsymbol{m}'$ and $\boldsymbol{X}'$ contain the updated representations for the query molecule and the support set molecules. The IterRefLSTM denotes the function which updates these representations. The main property of the IterRefLSTM module is that it is permutation-equivariant, thus a permutation $\pi(.)$ of the input elements results in the permutation of output elements: $\pi([\boldsymbol{m}', \boldsymbol{X}']) = \text{IterRefLSTM}_L(\pi([\boldsymbol{m}, \boldsymbol{X}]))$. Therefore, the full architecture is invariant to permutations of the support set elements. For details, we refer to Altae-Tran et al. (2017). The hyperparameter $L \in \mathbb{N}$ controls the number of iteration steps of the IterRefLSTM.

The IterRefLSTM also includes a similarity module which computes the predictions based on the updated representations mentioned above:

$$\boldsymbol{a} = \text{softmax}\left(\boldsymbol{k}\left(\boldsymbol{m}', \boldsymbol{X}'\right)\right)$$

$$\hat{y} = \sum_{n=1}^{N} a_n \, y_n,$$

where $\hat{y}$ is the prediction for the query molecule. For the computation of the attention values $\boldsymbol{a}$, the softmax function is used. $\boldsymbol{k}$ is a similarity metric, such as the cosine similarity.

**Hyperparameter search.** All hyperparameters were selected based on manual tuning on the validation set. We report the explored hyperparameter space in Table A3. Bold values indicate the selected hyperparameters for the final model.

### A.1.6 MHNFS: DETAILS AND HYPERPARAMETERS

The MHNfs consists of a molecule encoder, the context module, the cross-attention-module, and the similarity module. The molecule encoder is a fully-connected Neural Network, consisting of one layer with 1024 units. For the context module, a Hopfield layer with 8 heads is used and also the cross-attention module include 8 heads. We chose a concatenation of ECFPs and RDKit-based descriptors as the inputs for the MHNfs model. Notably, the RDKit-based descriptors were pre-processed in a way that instead of raw values quantils, which were computed by comparing a raw value with the

Table A4: Hyperparameter space considered for the MHNfs model selection. The hyperparameters of the best configuration are marked bold.

| Hyperparameter | Explored values |
|---|---|
| Molecule encoder | |
| • Number of hidden layers | **0**, 1, 2, 4 |
| • Number of units per hidden layer | **1024**, 4096 |
| • Output dimension | **512**, 1024 |
| • Activation function | ReLU, **SELU** |
| • Input dropout | **0.1** |
| • Dropout | **0.5** |
| Context module (hopfield layer) | |
| • Heads | **8**, 16 |
| • Association space dimension | 512 [512;2048] |
| • Dropout | 0.1, **0.5** |
| Cross-attention module (transformer mechanism) | |
| • Heads | 1, **8**, 10, 16, 32, 64 |
| • Number units in the hidden feedforward layer | **567** [512; 4096] |
| • Association space dimension | 1088 [512;2048] |
| • Dropout | 0.1, **0.5**, 0.6, 0.7 |
| • Number of layers: | **1**, 2, 3 |
| Similarity module: | |
| • Metric | cosine similarity, **dot product**, MinMax similarity |
| • Similarity space dimension | 512, **1024** |
| • $\tau$ | 32 [20;45] |
| Layer-normalization | False, **True** |
| • Affine | **False**, True |
| Training | |
| • Learning rate | **0.0001**, 0.001, 0.01 |
| • Optimizer | **Adam**, AdamW |
| • Weight decay | **0**, 0.0001 |
| • Batch size | **4096** |
| • Warm-up phase (epochs) | **5** |
| • Constant learning rate phase (epochs) | 25, **35** |
| • Decay rate | 0.994 |
| • Max. number of epochs | **350** |

distributation of all FS-Mol training molecules, were used. All descriptors were normalized based on the FS-Mol training data.

**Hyperparameter search.**  All hyperparameters were selected based on manual tuning on the validation set. We report the explored hyperparameter space in Table A4. Bold values indicate the selected hyperparameters for the final model. Early stopping points for the different reruns are chosen based on the $\Delta$AUC-PR metric on the validation set. For the five reruns the early-stopping points, that were automatically chosen by their validation metrics, were the checkpoints at epoch 94, 192, 253, 253 and 309.

**Model training.**  Figure A2 shows the learning curve of an exemplary training run of a MHNfs model on FS-Mol. The left plot shows the loss on the training set and the right plot shows the validation loss. The dashed line indicates the checkpoint of the model which was saved and then used for inference on the test set, whereas the stopping point was evaluated maximizing the $\Delta$AUC-PR metric on the validation set.

**Performance improvements in comparison to a naive baseline.**  Figure A3 shows a task-wise performance comparison between MHNfs and the Frequent Hitter model. Each point indicates a task in the test set and is colored according to their super-class membership. In 132 cases the MHNfs outperforms the frequent hitter model. In 25 cases the frequent hitter model yields better performance.

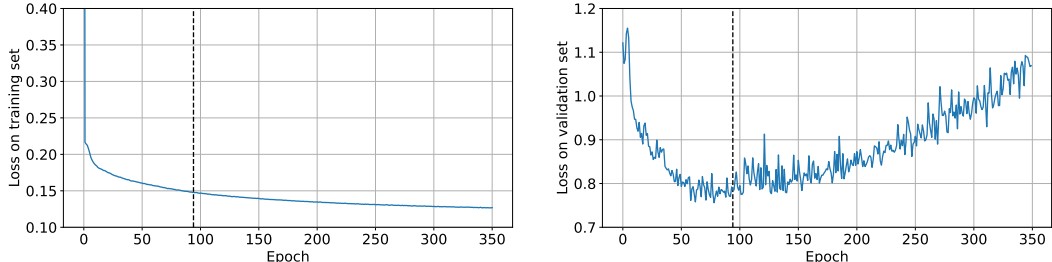

Figure A2: Exemplary MHNfs learning curve on FS-Mol. On the x-axis the number of epochs is displayed and on the y-axis the training loss (left) and the validation loss (right) is shown. The dashed line indicates the determined early-stopping point which is determined based on ΔAUC-PR on the validation set.

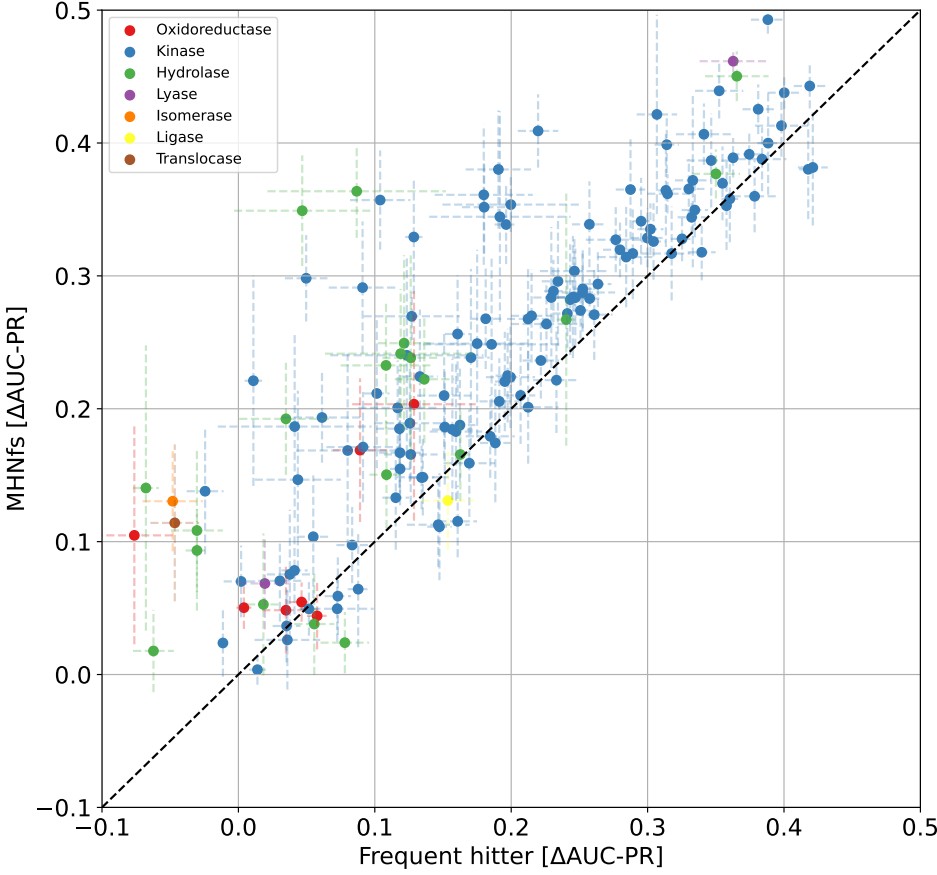

Figure A3: Performance comparison of MHNfs with the frequent hitter model. Each point refers to a task in the test set. Dashed lines indicate variablility across training reruns and different test support sets. The most points are located above the dashed line, which indicates that MHNfs performs better than den FH baseline at this task.

Table A5: Hyperparameter space considered for the PAR model selection. The hyperparameters of the best configuration are marked bold.

| Hyperparameter | Explored values |
| --- | --- |
| **Training** | |
| • Meta learning rate | $1.0 \cdot 10^{-05}$, $\mathbf{1.0 \cdot 10^{-04}}$, $1.0 \cdot 10^{-03}$, $1.0 \cdot 10^{-02}$ |
| • Inner learning rate | 0.01, **0.1** |
| • Update step | **1**, 2 |
| • Update step test | **1**, 2 |
| • Weight decay | $\mathbf{5.0 \cdot 10^{-05}}$, $1.0 \cdot 10^{-03}$ |
| • Epochs | 200000 |
| • Eval. steps | 2000 |
| **Encoder** | |
| • Use pre-trained GNN | **yes**, no |
| **Attention-based module** | |
| • Map dimension | 128, **512** |
| • Map layer | **2**, 3 |
| • Pre fc layer | **0**, 2 |
| • Map dropout | **0.1**, 0.5 |
| • Context layer | **2**, 3, 4 |
| **Relation graph** | |
| • Hidden dimension | 8, 128, **512** |
| • Number of layers | 2, **4** |
| • Number of layers for relation edge update | 2, **3** |
| • Batch norm | yes, **no** |
| • Relation dropout 1 | **0**, 0.25, 0.5 |
| • Relation dropout 2 | **0.2**, 0.25, 0.5 |

### A.1.7 PAR: DETAILS AND HYPERPARAMETERS

The PAR model (Wang et al., 2021) includes a pre-trained GNN encoder, which creates initial embeddings for the query and support set molecules. These embeddings are fed into an attention mechanism module which also uses activity information of the support set molecules to create enriched representations. Another GNN learns relations between query and support set molecules.

**Hyperparameter search.** For details we refer to Wang et al. (2021) and `https://github.com/tata1661/PAR-NeurIPS21/blob/main/parser.py`. All hyperparameters were selected based on manual tuning on the validation set. The hyperparameter choice for Tox21 (Wang et al., 2021) was used as a starting point. We report the explored hyperparameter space in Table A5. Bold values indicate the selected hyperparameters for the final model. Notably, we just report hyperparameter choices which were different from standard choices. We used a training script provided by (Wang et al., 2021), which can be found here `https://github.com/tata1661/PAR-NeurIPS21`.

### A.2 DETAILS ON THE FS-MOL BENCHMARKING EXPERIMENT

This section provides additional information for the FS-Mol benchmarking experiment (see Section 5).

**Memory-based baselines.** The Classic Similarity Search can be considered as a method with associative memory, where the label is retrieved from the memory. Notably, for this method, the associative memory is very limited since it is the support set. Siamese Networks, analogously to the Classic Similarity Search, retrieve the label from a memory, whereby the similarities are determined in a learned space. Also, the IterRefLSTM-based method can be seen as having a memory, whereby the LSTM controls storing and removing information from the training data by the input and the forget gate. In NLP, kNN-type memories are currently used. Conceptually, they are very similar to the Modern Hopfield Networks, setting the number of heads to one and choosing a suitable value for $\beta$.

Table A6: Results on FS-Mol [$\Delta$AUC-PR ]. The error bars represent standard deviation across training reruns.

| Method | $\Delta$AUC-PR |
|---|---|
| ADKF-IFT (Chen et al., 2022) | $.234 \pm .001$ |
| IterRefLSTM (Altae-Tran et al., 2017) | $.234 \pm .002$ |
| MHNfs | $.241 \pm .005$ |

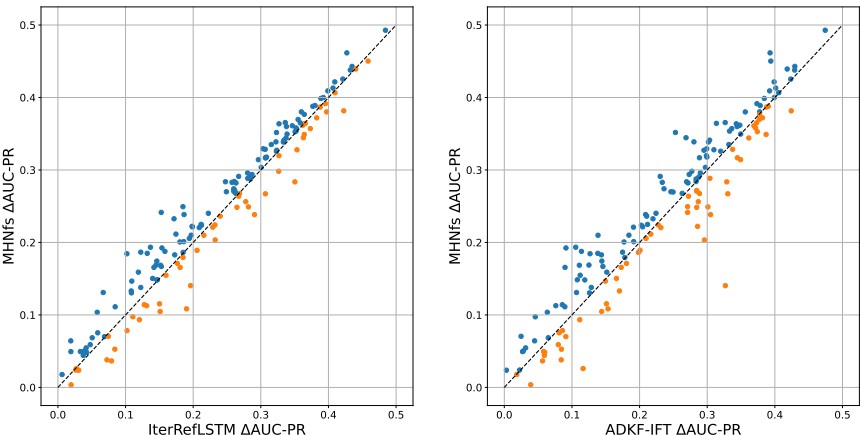

Figure A4: Task-wise model comparison. The left scatterplot shows a comparison between MHNfs and the IterRefLSTM-based method and the right scatterplot shows a comparison between MHNfs and ADKF-IFT. Each dot refers to a task in the test set. For tasks on which the MHNfs performs better related dots are colored blue; otherwise the dots are colored orange.

**Results.** The reported performance metrics comprise three different sources of variation, namely variation across different tasks, variation across different support sets during test time, and variation across different training reruns. While error bars in Table 1 report variation across tasks, error bars in Table A6 report variation across training reruns. For ADKF-IFT, the authors provided error bars for every single test task. Based on these error bars we sampled performance values to be able to compare ADKF-IFT with the MHNfs training reruns. Figure A4 shows a task-wise model comparison between a) MHNfs and the IterRefLSTM-based method and b) MHNfs and ADKF-IFT. For a) MHNfs performs better on 106 of 157 tasks and therefore significantly outperforms the IterRefLSTM-based method (binomial test $p$-value $6.8 \cdot 10^{-6}$). For b) MHNfs performs better on 98 tasks and therefore significantly outperforms ADKF-IFT (binomial test $p$-value $0.001$), too. Notably, ADKF-IFT performs better on non kinase-targets which can be seen in Table 1.

## A.3 DETAILS ON THE ABLATION STUDY

The MHNfs has two new main elements compared to the most similar previous state-of-the art method IterRefLSTM, which are the context module and the cross-attention-module. In this ablation study we aim to investigate i) the importance of all design elements, which are the context module, the cross-attention module, and the similarity module, and ii) the superiority of the cross-attention module compared to the IterRefLSTM module.

### A.3.1 ABLATION STUDY A: COMPARISON AGAINST ITERREFLSTM

For a fair comparison between the cross-attention module and the IterRefLSTM we used a pruned MHN version ("MHNfs -CM") which has no context module and compared it with the IterRefLSTM model. The evaluation includes five training reruns each and ten different support set samplings.

Table A7: Results of the ablation study on FS-Mol [AUC, $\Delta$AUC-PR ]. The error bars represent standard deviation across training reruns and draws of support sets. The $p$-values indicate whether the difference between two models in consecutive rows is significant.

| Method | AUC | $\Delta$AUC-PR | $p_{\mathrm{AUC}}$[a] | $p_{\Delta\mathrm{AUC-PR}}$[a] |
|---|---|---|---|---|
| MHNfs (CM+CAM+SM) | $.739 \pm .005$ | $.241 \pm .006$ | | |
| MHNfs -CM | $.737 \pm .004$ | $.240 \pm .005$ | 0.030 | 0.002 |
| MHNfs -CM -CAM | $.719 \pm .006$ | $.223 \pm .006$ | < 1.0e-8 | <1.0e-8 |
| Similarity Search | $.604 \pm .003$ | $.113 \pm .004$ | <1.0e-8 | < 1.0e-8 |
| IterRefLSTM (Altae-Tran et al., 2017)[b] | $.730 \pm .005$ | $.234 \pm .005$ | <1.0e-8 | 8.73e-7 |

[a] paired Wilcoxon rank sum test  [b] IterRefLSTM is compared to MHNfs -CM

The results, reported as the mean across training reruns and support sets, can be seen in Table A7. We performed a paired Wilcoxon rank sum test for both the AUC and the $\Delta$AUC-PR metric. Both $p$-values indicate high significance.

### A.3.2 ABLATION STUDY B: ALL DESIGN ELEMENTS

We evaluate the performance of all main elements within the MHNfs, which are the context module, the cross-attention module, the similarity module and the molecule encoder. For this analysis, we start with the complete MHNfs which includes all modules and report AUC and $\Delta$AUC-PR performance values. Then, we iteratively omit the individual modules, measuring whether there is a significant performance difference with and without the module. Table A7 shows the results, where performance values for the full MHNfs, a MHNfs model without the context module ("MHNfs -CM") and a MHNfs module without the context and the cross-attenion module ("MHNfs -CM -CAM") is included. Notably, the model without the context module and without the cross-attention module just consists of a learned molecule encoder and the similarity module. We evaluted the impact of the learned molecule encoder by replacing it with a fixed encoder, which maps a molecule to its descriptors. The model with the fixed encoder is a classic chemoinformatics method which is called Similarity Search (Cereto-Massagué et al., 2015).

For the evaluation, we performed five training reruns for every model and sampled ten different support sets for every task. Table A7 shows the results in terms of AUC and $\Delta$AUC-PR. We performed paired Wilcoxon rank sum tests on both metrics, comparing two methods in consecutive rows in the table. The table shows that every module has a significant impact as omitting a module results in a significant performance drop. The comparison between the MHNfs version without the context module and without the cross-attention module with the Similarity Search showed a significant superiority of the learned molecule encoder in comparison to the fixed encoder.

### A.3.3 ABLATION STUDY C: UNDER DOMAIN SHIFT ON TOX21

Referring to Section A.3.2, the context module and the cross-attention module showed their importance for the global architecture. This importance gets even more pronounced for the domain shift experiment on Tox21 as one can see in Table A8.

Again, five training reruns and ten support set draws are used for evaluation. Including the context module makes a clear and significant difference for both metrics AUC and $\Delta$AUC-PR.

### A.4 DETAILS ON THE DOMAIN SHIFT EXPERIMENTS

This section provides additional information for the Domain shift experminet on Tox21.

**Results.** The reported performance metrics comprise three different sources of variation, namely variation across different tasks, variation across different support sets during test time, and variation across different training reruns. While error bars in Table 2 report variation across both, drawn support sets and training reruns, error bars in Table A9 just report variation across training reruns. Notably, for the Similarity Search, the performance values do not vary since the model does not include any trainable parameters.

Table A8: Results of the ablation study on Tox21 [AUC, $\Delta$AUC-PR ]. The error bars represent standard deviation across training reruns and draws of support sets. The $p$-values indicate whether a model is significantly different to the MHNfs in terms of the AUC and $\Delta$AUC-PR metric.

| Method | AUC | $\Delta$AUC-PR | $p_{\text{AUC}}$[a] | $p_{\Delta\text{AUC}-\text{PR}}$[a] |
|---|---|---|---|---|
| MHNfs (CM+CAM+SM) | .679 $\pm$ .018 | .073 $\pm$ .008 | | |
| MHNfs -CM | .662 $\pm$ .028 | .069 $\pm$ .012 | 6.28e-8 | 0.002 |
| MHNfs -CM -CAM | .640 $\pm$ .018 | .057 $\pm$ .009 | <1.0e-8 | <1.0e-8 |
| Similarity Search | .629 $\pm$ .015 | .061 $\pm$ .008 | <1.0e-8 | <1.0e-8 |
| IterRefLSTM | .664 $\pm$ .018 | .067 $\pm$ .008 | 2.53e-6 | 3.38e-5 |

[a] paired Wilcoxon rank sum test

Table A9: Results of the domain shift experiment on the Tox21 dataset [AUC, $\Delta$AUC-PR]. The best method is marked bold. Error bars represent standard deviation across training reruns

| Method | AUC | $\Delta$AUC-PR |
|---|---|---|
| Similarity Search (baseline)[a] | .629 $\pm$ .000 | .061 $\pm$ .000 |
| IterRefLSTM (Altae-Tran et al., 2017) | .664 $\pm$ .004 | .067 $\pm$ .001 |
| MHNfs | **.679** $\pm$ .009 | **.073** $\pm$ .003 |

[a] The Similarity Search does not include any learned parameters. Therefore, there is no variability across training reruns.

## A.5 GENERALIZATION TO DIFFERENT SUPPORT SET SIZES

In the following section, we test the ability of MHNfs to generalize to different support set sizes. During training in the FS-Mol benchmarking setting, the MHNfs model has access to support sets of size 16. However, at inference, the support set size might be different. Figure A5 provides performance estimates of the support-set-size-16 MHNfs models on other support set sizes. Note that the estimates could be seen as approximate lower bounds of the predictive performance on settings with different support set sizes (y-axis labels). For a model used in production or in a real-world drug discovery setting, MHNfs should be trained with varying support set sizes that resemble the distribution of real drug discovery projects.

Triantafillou et al. (2019) analysed the performance of different few-shot models across different support set sizes. Their analysis showed that in very-low-data settings embedding-based methods, namely Prototypical Networks and fo-Proto-MAML, performed best. In contrast to that, finetuning-based method significantly profit from larger support set sizes (Triantafillou et al., 2019).

MHNfs is an embedding-based method and performs – in accordance with the findings mentioned above (Triantafillou et al., 2019) – well for small support set sizes (see Table 1). Following Triantafillou et al. (2019), it is exactly the settings related to these smaller support set sizes, e.g. a support set size of 16, which are suitable for MHNfs. For large support set sizes, e.g. 64 or 128, we point to the work done by Chen et al. (2022), in which the fine-tuning method ADKF-IFT achives an $\Delta$AUC-PR-score $> 0.3$ for large support set sizes.

## A.6 GENERALIZATION TO DIFFERENT CONTEXT SETS

In this section, we test the ability of MHNfs to generalize to different context sets. While the FS-Mol training split is used as a context during training, we assessed whether our model is robust to different context sets for inference. To this end we preprocessed the GEOM dataset (Axelrod and Gomez-Bombarelli, 2022) from which we used 100,000 molecules that passed all pre-processing checks. From this set, we sample 10,000 molecules as context set for MHNfs. Because GEOM contains drug-like molecules, similar to FS-Mol the predictive performance remains stable (see Table A10).

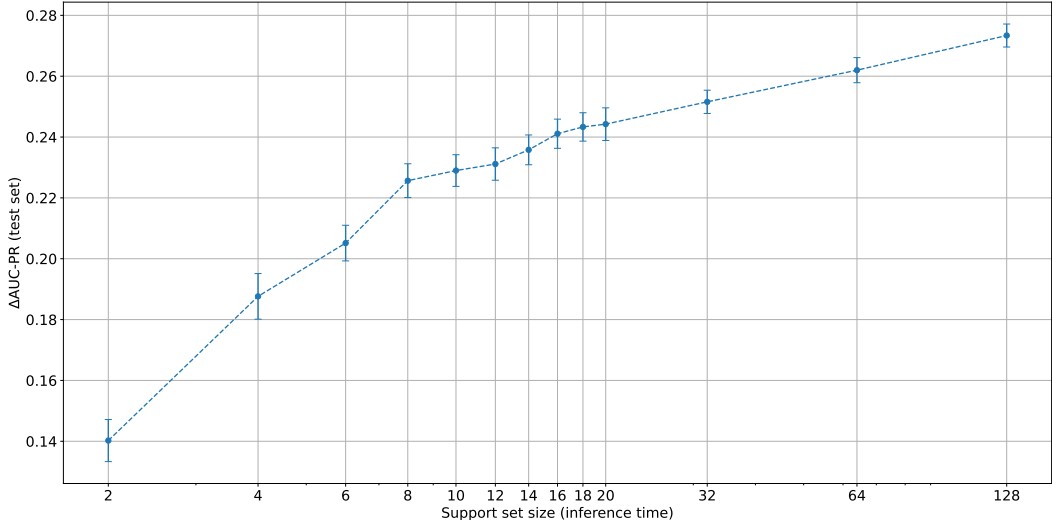

Figure A5: Performance of MHNfs for different support set sizes during inference time. The MHNfs models are trained with support sets of the size 16.

Table A10: MHNfs performance for different context sets [$\Delta$AUC-PR ]. The error bars represent standard deviation across training reruns and draws of support sets.

| Dataset used as a context | $\Delta$AUC-PR |
|---|---|
| FS-Mol (Stanley et al., 2021) | .2414 ± .006 |
| GEOM (Axelrod and Gomez-Bombarelli, 2022) | .2415 ± .005 |

### A.7 Details and insights on the context module

The context module replaces the initial representations of query and support set molecules by a retrieval from the context set. The context set is a large set of molecules and covers a large chemical space. The context module learns how to replace the initial molecule embeddings such that the context-enriched representations are put in relation to this large chemical space and still contains all necessary information for the similarity-based prediction part. Figure A6 shows the effect of the context module for the MHNfs model. Extreme initial embeddings, such as the purple embedding on the right, are pulled more into the known chemical space, represented by the context molecules. Notably, the replacement described above is a soft replacement, because also the initial embeddings contribute to the context-enriched representations due to skip-connections.

### A.8 Reinforcing the covariance structure in the data using modern Hopfield networks

We follow the argumentation of (Fürst et al., 2022, Theorem A3) that retrieval from an associative memory of a MHN reinforces the covariance structure.

Let us assume that we have one molecule embedding from the query set $m \in \mathbb{R}^d$ and one molecule embedding from the support set $x \in \mathbb{R}^d$ and both have been enriched with the context module with memory $C \in \mathbb{R}^{d \times M}$ (ignoring linear mappings):

$$m' = C \operatorname{softmax}(\beta C^T m) \tag{A9}$$

$$x' = C \operatorname{softmax}(\beta C^T x) \tag{A10}$$

Then the similarity of the retrieved representations as measured by the dot product can be expressed in terms of covariances:

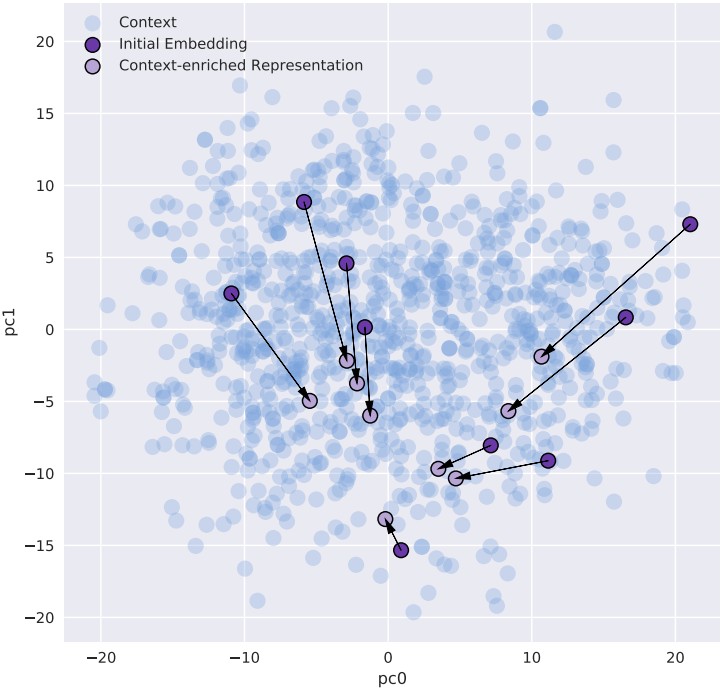

Figure A6: PCA-downprojection plot of molecule embeddings. Each dot represents a molecule embedding, of which the first two components are displayed on the x- and y-axis. Blue dots represent context molecules. Dark purple dots represent initial embeddings for some exemplary molecules, of which some exhibit extreme characteristics and are thus located away from the center. Arrows and light purple dots represent the enriched molecule embeddings after the retrieval step. Especially molecules from extreme positions are moved stronger to the center and thus are more similar to known molecules after retrieval.

$$\boldsymbol{m}'^{T}\boldsymbol{x}' = \mathrm{softmax}(\beta\boldsymbol{C}^{T}\boldsymbol{m})^{T}\boldsymbol{C}^{T}\boldsymbol{C}\mathrm{softmax}(\beta\boldsymbol{C}^{T}\boldsymbol{x}) = \tag{A11}$$

$$= (\bar{\boldsymbol{c}} + \mathrm{Cov}(\boldsymbol{C},\boldsymbol{m})^{T}\boldsymbol{m})^{T}\,(\bar{\boldsymbol{c}} + \mathrm{Cov}(\boldsymbol{C},\boldsymbol{x})\boldsymbol{x}), \tag{A12}$$

where $\bar{\boldsymbol{c}}$ is the row mean of $\boldsymbol{C}$ and following the *weighted covariances* are used:

$$\mathrm{Cov}(\boldsymbol{C},\boldsymbol{m}) = \boldsymbol{C}\mathrm{J}^{\mathrm{m}}(\beta\boldsymbol{C}\boldsymbol{m})\boldsymbol{C}^{T} \qquad \mathrm{Cov}(\boldsymbol{C},\boldsymbol{x}) = \boldsymbol{C}\mathrm{J}^{\mathrm{m}}(\beta\boldsymbol{C}\boldsymbol{x})\boldsymbol{C}^{T}. \tag{A13}$$

$\mathrm{J}^{\mathrm{m}} : \mathbb{R}^{M} \mapsto \mathbb{R}^{M \times M}$ is a mean Jacobian function of the softmax (Fürst et al., 2022, Eq.(A172)).

The Jacobian J of $\boldsymbol{p} = \mathrm{softmax}(\beta\boldsymbol{a})$ is $\mathrm{J}(\beta\boldsymbol{a}) = \beta\,\big(\mathrm{diag}(\boldsymbol{p}) - \boldsymbol{p}\boldsymbol{p}^{T}\big).$

$$\boldsymbol{b}^{T}\mathrm{J}(\beta\boldsymbol{a})\,\boldsymbol{b} \;=\; \beta\,\boldsymbol{b}^{T}\,\big(\mathrm{diag}(\boldsymbol{p}) \;-\; \boldsymbol{p}\,\boldsymbol{p}^{T}\big)\,\boldsymbol{b} \;=\; \beta\,\left(\sum_{i}p_{i}\,b_{i}^{2} \;-\; \left(\sum_{i}p_{i}\,b_{i}\right)^{2}\right), \tag{A14}$$

this is the second moment minus the mean squared, which is the variance. Therefore, $\boldsymbol{b}^{T}\mathrm{J}(\beta\boldsymbol{a})\boldsymbol{b}$ is $\beta$ times the covariance of $\boldsymbol{b}$ if component $i$ is drawn with probability $p_{i}$ of the multinomial distribution $\boldsymbol{p}$. In our case the component $i$ is context sample $\boldsymbol{c}_{i}$. $\mathrm{J}^{\mathrm{m}}$ is the average of $\mathrm{J}(\lambda\boldsymbol{a})$ over $\lambda = 0$ to $\lambda = \beta$.

Note that we can express the enriched representations using these covariance functions:

$$\boldsymbol{m}' = (\bar{\boldsymbol{c}} + \mathrm{Cov}(\boldsymbol{C},\boldsymbol{m})^{T}\boldsymbol{m}) \tag{A15}$$

$$\boldsymbol{x}' = (\bar{\boldsymbol{c}} + \mathrm{Cov}(\boldsymbol{C},\boldsymbol{x})^{T}\boldsymbol{x}), \tag{A16}$$

which connects retrieval from MHNs with reinforcing the covariance structure of the data.

### A.9 Discussion, limitations and broader inpact

In a benchmarking experiment, the architecture was assessed for its ability to learn accurate predictive models from small sets of labelled molecules and in this setting it outperformed all other methods. In a domain shift study, the robustness and transferability of the learned models has been assessed and again **MHNfs** exhibited the best performance. The resulting predictive models often reach an AUC larger than .70, which means that enrichment of active molecules is expected (Simm et al., 2018) when the models are used for virtual screening. It has not escaped our notice that the specific context module we have proposed could immediately be used for few-shot learning tasks in computer vision, but might be hampered by computational constraints.

Effectively using the information stored in the training data for new tasks is not only a key for our context-module but also for a lot of other few-shot strategies like pre-training or meta-learning. For pre-training and meta-learning based approaches, this information is stored in the model weights, while the context module directly has access to it via an external memory. We believe that accessing this information directly via an external memory is benefitial in this setting because a) pre-training for small molecule drug discovery is a promising approach, but still comes with its own challenges (Xia et al., 2022) and b) a meta-learning approach, like MAML, needs labeled data while Modern Hopfield Networks operate on unlabeled data and therefore might be able to give access to more comprehensive information in the data including unlabeled data points.

**Limitations.** In the FS-Mol benchmark experiment, the runner-up method ADKF-IFT (Chen et al., 2022) performed better on non kinase-tasks. We hypothesize that we could improve the MHNfs performance for non kinase tasks by upsampling the other task sub-groups. While the implementation of our method is currently limited to small, organic drug-like molecules as inputs, our conceptual approach can also be used for macro-molecules such as RNA, DNA or proteins. The output domain of our method comprises biological effects, such that the prediction must be understood in that domain. Our method demands higher computational costs and memory footprint as other embedding-based methods because of the calculations necessary for the context module. While we hypothesize that our approach could also be successful for similar data in the materials science domain, this has not been assessed. Our study is also constrained by a limited amount of hyperparameter search for all methods. Deep learning methods usually have a large number of hyperparameters, such as hidden dimensions, number of layers, learning rates, of which we were only able to explore the most important ones. The composition and choice of the context set is also under-explored and might be improved by selecting reference molecules with an appropriate strategy.

**Broader impact.** *Impact on machine learning and related scientific fields.* We envision that with (a) the increasing availability of drug discovery and material science datasets, (b) further improved biotechnologies, and (c) accounting for characteristics of individuals, the drug and materials discovery process will be made more efficient. For machine learning and artificial intelligence, the novel way in which representations are enriched with context might strengthen the general research stream to include more context into deep learning systems. Our approach also shows that such a system is more robust against domain shifts, which could be a step towards Broad AI (Chollet, 2019; Hochreiter, 2022). *Impact on society.* If the approach proves useful, it could lead to a faster and more cost-efficient drug discovery process. Especially the COVID-19 pandemic has shown that it is crucial for humanity to speed up the drug discovery process to few years or even months. We hope that this work contributes to this effort and eventually leads to safer drugs developed faster. *Consequences of failures of the method.* As common with methods in machine learning, potential danger lies in the possibility that users rely too much on our new approach and use it without reflecting on the outcomes. Failures of the proposed method would lead to unsuccessful wet lab validation and negative wet lab tests. Since the proposed algorithm does not directly suggest treatment or therapy, human beings are not directly at risk of being treated with a harmful therapy. Wet lab and in-vitro testing would indicate wrong decisions by the system. *Leveraging of biases in the data and potential discrimination.* As for almost all machine learning methods, confounding factors, lab or batch effects, could be used for classification. This might lead to biases in predictions or uneven predictive performance across different drug targets or bioassays.

