# OpenReview forum: "Context-enriched molecule representations improve few-shot drug discovery"
_ICLR.cc/2023/Conference — ICLR 2023 poster_

### Official Review · Reviewer_kPjE · 2022-10-24

**Confidence:** 4
**Correctness:** 4
**Technical Novelty And Significance:** 3
**Empirical Novelty And Significance:** 3
**Recommendation:** 8

**Clarity, Quality, Novelty And Reproducibility:**

This work is novel and appears to be of high quality. In particular, a large number of methods are included for comparison, including new rational baselines for the FS-MOL tasks, and descriptions of each method and how they were trained are provided, though no code is provided. The authors include additional robustness results for MHNfs in the appendix.

The paper is written very clearly, and the work is original due to the use of associated memory (MHNs) for contextual enrichment.

**Strength And Weaknesses:**

Strengths:
- Proposes a novel few-shot mechanism based on comparisons to knowns molecules, which is loosely analogous to how medicinal chemists typically explore chemical space.
- Demonstrates that the novel mechanism outperforms all models to which it was compared.
- Shows that a very simple predictor based on average activity performs better than most previous few-shot learning methods.

Weaknesses:
- The performance of MHNfs on FS-MOL tasks and when generalizing to Tox21 is within the standard error of the next-best model. Moreover, MHNfs perform worse on non-kinase-related tasks. The authors should point these facts out, perhaps addressing the contributions of different factors to the standard errors.
- No other comparable memory-based model is included for comparison. Comparison to a simple memory-based baseline would help understand the role of memory in the reported performance improvement.


**Summary Of The Paper:**

To improve upon the current best performance of models that predict molecular properties in typical drug discovery scenarios (where the number of molecules with known properties are very small), the authors propose MHNfs, an embedding-based few-shot learner using Modern Hopfield Networks to provide learned associative memory for enriching a given molecular representation with those of reference molecules, in order that the properties of the given molecule are more accurately predicted. The authors show that 1) MHNfs outperform a variety of other few-shot learning methods on the activity prediction tasks in the FS-MOL dataset, and 2) that contextual enrichment is responsible for the improvement in performance over the next-best method. They also demonstrate that their method generalizes better to non-drug-like molecules than the next-best method.

**Summary Of The Review:**

The use of associative memory for enriching molecular representations is novel, and the improvement of MHNfs on FS-MOL tasks over other methods appears significant, despite being within the standard error of the next-best method. The establishment of simple, rational baselines for the FS-MOL tasks is also an important contribution to understanding that benchmark and the source of performance gains (or losses).

---

> ### Author Response · Authors · 2022-11-15
> **Answer to reviewer kPjE**
>
> We thank the reviewer for their valuable feedback which helps a lot to polish our manuscript. We are glad that the reviewer assessed our approach to be novel and our method to outperform the others. Also, we are glad the reviewer likes our Frequent Hitter baseline.
>
> ### Performance values
> As indicated in the general response (see above), the error bars are overlapping because they contain multiple sources of variation, of which the strongest one is the variation across tasks. If this variation is averaged out, the improvement by our method becomes evident. We provide more information on that in the new version of the manuscript and apologize that we were not able to bring across the crucial information about the error bars of the FS-Mol and Tox21 benchmarks. We added the requested information in Section A.2 (p. 23-24), Section A.4 (p. 25-26) and Section A.9 (p. 30).
>
>
> ### Memory-based baselines
> We apologize we have not made connections clearer. We added  information in this regard in Section A2 (p. 23). The Classic Similarity Search can be considered as a method with associative memory, where the label is retrieved from the memory. Notably, for this method, the associative memory is very limited since it is the support set. Siamese Networks, analogously to the Classic Similarity Search, retrieve the label from a memory, whereby the similarities are determined in a learned space. Also, the IterRefLSTM based method can be seen as having a memory, whereby the LSTM controls storing and removing information from the training data by the input and the forget gate. In NLP, kNN-type memories are currently used. Conceptually, they are very similar to the Modern Hopfield Networks, setting the number of heads to one and choosing a suitable value for $\beta$.

---

### Official Review · Reviewer_XyoZ · 2022-10-24

**Confidence:** 4
**Correctness:** 4
**Technical Novelty And Significance:** 3
**Empirical Novelty And Significance:** 3
**Recommendation:** 8

**Clarity, Quality, Novelty And Reproducibility:**

Clarity:
The work is very clear and well-written.
Quality:
The work is of clearly high standard and appropriately addresses all baselines thoroughly.
Novelty:
While many of the components have previously been applied in few-shot learning, and MHNs are of course not new, this combination of components and insight into its value in the few-shot molecular property prediction are novel and useful.

**Strength And Weaknesses:**

Strengths
* The paper is very clearly written and thorough; the model is transparently described. While a relatively simple idea, the execution is excellent and baseline comparisons thorough.
* The ablation studies are informative.
* The novel method proposed works well: the performance gains are significant in a challenging setting
* The paper also addresses in detail a number of state-of-the-art baselines, and importantly brings to the attention of the machine learning community the importance of the Frequent Hitters naive baseline.
* The appendices are similarly thorough.

Weaknesses
* It would be somewhat valuable to have more discussion around why the method is regularizing the covariance as the authors discuss and how this compares to/ why this is outperforming a pretraining or meta-learning strategy using the entire training set of tasks available in FS-Mol.


**Summary Of The Paper:**

This paper addresses molecular property prediction in the few-shot setting by suggesting a novel Modern Hopfield Network architecture and testing it on the molecule-specific few-shot dataset FS-Mol. The novel architecture seeks to make use of the context available in the wider molecular training set through means other than a pretraining step.

**Summary Of The Review:**

This is a simple but effective idea that beats previously suggested methods on this task. While the idea is simple, the work is well executed, described, and thoroughly explored, and thus makes a valuable contribution to this particular body of work.

---

> ### Author Response · Authors · 2022-11-15
> **Answer to reviewer XyoZ**
>
> We thank the reviewer for all the work and valuable feedback which helps to enhance our manuscript. We are glad the author assessed our work to be clearly written and thorough, our execution to be excellent and the performance gains to be significant. Also, we are pleased that the author finds our ablation study to be informative and appreciates our baselines.
>
> ### Extended discussion around why the method is regularizing the covariance and why this is outperforming pre-training
> Modern Hopfield networks amplify co-occurrences and the covariance structure. Replacing the original embeddings by retrieved representations reinforces features that frequently occur together in stored embeddings. Additionally, spurious co-occurrences that are peculiar to a sample are averaged out. By this means, the covariance structure is reinforced by the retrieval as shown in Section A8 (p. 28-29). Section A7 and Figure A5 describe and show the effect of the context module. We will add a more extensive discussion in the final version of the manuscript.
> Effectively using the information stored in the training data for new tasks, is not only a key for our context-module but also for a lot of other few-shot strategies like pre-training or meta-learning. Therefore, the question which is raised here by the reviewer is a very good one, but still difficult to answer. A precise answer appears difficult because pre-training and meta-learning on the one hand and our retrieval-based approach on the other hand follow quite different strategies. While, for pre-training and meta-learning based approaches, the information available in the dataset is stored in the model weights, the context module directly has access to this information via an external memory.
> Nevertheless, we try to give an answer in the following manner:
> - Pre-training for small molecule drug discovery is a promising approach, but still comes with its own challenges [1].
> - A meta-learning approach, like MAML, needs labeled data. This might be the important difference since the Modern Hopfield Network operates on unlabeled data and therefore might be able to give access to more comprehensive information in the data including unlabeled data points.
> - Apart from ADKF-IFT, which partly relies on meta-learning, the FS-Mol benchmark experiment shows that it is rather the embedding-based and not the optimizer-based methods which perform well.
> - In computer vision tasks with very few data points, embedding-based methods also outperform optimizer-based approaches, which we hypothesize may be due to the high variance of the optimizer steps [2]. This would mean that the critical step is to control the variance brought in by the support set and this is exactly what the context module does: it removes peculiarities and spurious co-occurrences by comparing with context.
>
> We will elaborate on these matters more in the Discussion Section in the final version of the manuscript.
>
> ### References.
> [1] Xia, Jun, et al. "A Systematic Survey of Molecular Pre-trained Models." arXiv preprint arXiv:2210.16484 (2022).
>
> [2] Triantafillou, E., Zhu, T., Dumoulin, V., Lamblin, P., Evci, U., Xu, K., Goroshin, R., Gelada, C., Swersky, K., Manzagol, P.-A., et al. (2019). Meta-dataset: A dataset of datasets for learning to learn from few examples. arXiv preprint arXiv:1903.03096.

---

> > ### Comment · Reviewer_XyoZ · 2022-11-27
> > **Response to authors**
> >
> > I thank the author for their response. The additional discussion around the impact of MHNs vs. pre-training or meta-learning strategies is careful and helpful for the field.

---

### Official Review · Reviewer_WgGe · 2022-10-25

**Confidence:** 4
**Correctness:** 3
**Technical Novelty And Significance:** 3
**Empirical Novelty And Significance:** 3
**Recommendation:** 6

**Clarity, Quality, Novelty And Reproducibility:**

Section 2 - I found the problem setting to be very clear.

The standard errors of models in Table 1 are overlapping. Is it possible to chose a single best performing model if these standard errors overlap? To what confidence does a given model/architecture perform over others?

Equations: X, X’, X’’, etc: Would it make more sense to have them as Z, since they are some embedding? It is unclear to me as a reader as how to track x, X, as well as any ‘ of X, when some are molecules, and others are embeddings or representations.

**Strength And Weaknesses:**

I really appreciate the implementation of the baselines, as well as the authors’ interpretation of them (Frequent Hitters and Similarity Search). Upon release of code, it is useful to see those implementations as well.

In Section 5.3 - A domain shift experiment is performed. There 8 positive and 8 negative were randomly selected. How different are they? Can better sets of divergent molecules be made by clustering based on molecular fingerprints and/or Tanimoto distances?

Section 5.3 - Are there better datasets than Tox21 for this analysis? It is difficult to really see your model outperform others in this challenge. Generally, I feel that this community is hitting diminishing returns with Tox21, so understanding if algorithmic improvements actually improve performance is challenging.


**Summary Of The Paper:**

The authors develop a deep learning method on molecules to better leverage training data and contextual information to make property predictions.

**Summary Of The Review:**

I found the work to be interesting and somewhat complex (albeit convoluted). Though the work was well presented and in a generally interesting research area, it feels like this work, as demonstrated in the paper, is hitting diminishing returns, and could find additional evidence to help bolster and argument for the utility and necessity of this approach.

---

> ### Author Response · Authors · 2022-11-15
> **Answer to reviewer WgGe - Part 2/2**
>
> ### Notation
> We thank the author for this valuable feedback. During the writing of the manuscript, we tested many different types of notation and found the notation with the primes x, x’, x’’, which indicate enriched versions of the same molecule, to be the least cluttered. We are sorry that, nevertheless, the author has some issues with it. Of course we are open to change and improve our notation. It first might be beneficial to share our thoughts about it though:
> - We denote the support set $\{(\boldsymbol x_1, y_1),\ldots,(\boldsymbol x_N,y_N) \}$ by $\boldsymbol Z$ (p. 16). This is consistent with other few-shot literature. Even though $\boldsymbol Z$ does not appear in the main paper, we would prefer to not denote molecule representations by $\boldsymbol z$ for consistency reasons throughout our manuscript.
> - We aimed for a self-explaining notation in a sense that the naming of the molecular representation already contains information about the original input and therefore links the representations which occur within our model naturally to its input —  i.e, the query or a support set molecule. To distinguish the input from a representation within our model we denoted the input query and support set molecules by $m$ and $x$, whereas the created representations within MHNfs are named by $\boldsymbol m$, $\boldsymbol m’$, $\boldsymbol m’’$, and $\boldsymbol x’$, $\boldsymbol x’’$, $\boldsymbol x’’$. Here:
>     - $m$ / $x$ describes a symbolic or low-level representation of the molecule as it is stored in the dataset — e.g., the SMILES string or the molecular graph.
>     - $\boldsymbol m$ / $\boldsymbol x$ is a vector.
> We hoped that equation 4 and 5 (page 4) help to introduce and understand our notation.
> With knowledge about our thoughts, we would highly appreciate getting more details about what causes the issues. If the notation with the prime is causing the problems,  we think that replacing x by z would not help either. One would have to come up with a new letter for each newly created representation which we thought easily leads to a very cluttered notation. In case it is the similarity between a symbolic or low-level representation of a molecule and the MHNfs-internal representations of that molecule which creates a problem we could rename $x$ by $\mathbb x$ and $m$ by $\mathbb m$.
>
> ### References
> [1] Alperstein, Z., Cherkasov, A., and Rolfe, J. T. (2019). All smiles variational autoencoder. arXiv
> preprint arXiv:1905.13343.
>
> [2] Klambauer, G., Unterthiner, T., Mayr, A., and Hochreiter, S. (2017). Self-normalizing neural networks. In Advances in neural information processing systems 30, pages 972–981
>
> [3] Duvenaud, D., Maclaurin, D., Aguilera-Iparraguirre, J., Gómez-Bombarelli, R., Hirzel, T., Aspuru-Guzik, A., and Adams, R. P. (2015). Convolutional networks on graphs for learning molecular fingerprints. arXiv preprint arXiv:1509.09292
>
> [4] Li, J., Cai, D., and He, X. (2017). Learning graph-level representation for drug discovery. arXiv preprint arXiv:1709.03741.
>
> [5] Li, P., Li, Y., Hsieh, C.-Y., Zhang, S., Liu, X., Liu, H., Song, S., and Yao, X. (2021). Trimnet: learning molecular representation from triplet messages for biomedicine. Briefings in Bioinformatics, 22(4):bbaa266.
>
> [6] Zaslavskiy, M., Jégou, S., Tramel, E. W., and Wainrib, G. (2019). Toxicblend: Virtual screening of toxic compounds with ensemble predictors. Computational Toxicology, 10:81–88.

---

> ### Author Response · Authors · 2022-11-15
> **Answer to reviewer WgGe - Part 1/2**
>
> We thank the reviewer for all the work and valuable feedback which helps to enhance our manuscript. We are glad the author assessed our work to be interesting and well presented. Also, we are glad the author appreciates our baselines and finds our problem setting to be described clearly.
>
> ### Error bars
> We thank the reviewer for bringing up this point. We improved our manuscript based on this input and provide more information in the general response (see general response above).
>
> ### Baselines
> We are glad the author likes the baselines we added to the FS-Mol benchmark results. We will publish our implementations for the baselines together with our final manuscript version.
>
> ### Tox21 - domain shift experiment
> We share the author’s opinion that generally our community is hitting diminishing returns with Tox21. One of the main reasons is that the originally defined compound-wise training-, validation-, and test-split is not respected. We respect the original split in a way that we draw the support sets from the original compound-wise training split whereas the test set is used as query molecules (Section 5.3, p.9, passage: Training and evaluation).
> We do not set a new state-of-the-art for Tox21 in the general many-shot setting. State-of-the-art supervised learning methods that have access to the full training set reach AUC performance values up to 0.871 [1-6]. We show that MHNfs exhibits robustness under a strong domain shift from drug-like molecules of FS-Mol to environmental chemicals, pesticides, food additives of Tox21. Additionally, there is a target shift from kinases, hydrolases, and oxidoreductases of FS-Mol to nuclear receptors and stress responses of Tox21. This is why the Tox21 dataset perfectly fits the needs of our domain-shift experiment.
> We added information about state-of-the art methods in the manuscript (Section 5.3, p. 9).
>
> ### Support set variability
> We did not perform a specific analysis on that. However, for both experiments, the FS-Mol benchmarking as well as for the Tox21 domain shift experiment, we randomly sampled support sets multiple times, and evaluated and reported model performances across these different support sets. Due to the random sampling procedure we hypothesize to have evaluated the models on both, more pair-wise similar and different support sets.

---

### Official Review · Reviewer_b2MR · 2022-10-25

**Confidence:** 4
**Correctness:** 3
**Technical Novelty And Significance:** 3
**Empirical Novelty And Significance:** 3
**Recommendation:** 6

**Clarity, Quality, Novelty And Reproducibility:**

Questions:
Are the context molecules for the Tox21 experiment sampled from the Tox21 or FS-MOL pretraining dataset and how many were used?
How much is the performance benefit dependent on the size of the context, in the FS-MOL experiments, 5% of training samples were randomly sampled. Have other % attempted?
What is the extent of computational burden from the context on the inference?
Could the proposed approach be applied to other few-shot learning domains outside molecular learning?


**Strength And Weaknesses:**

Strengths:
Competitive few-shot learning performances
The approach to use the context of other training samples is novel and interesting.

Weaknesses:
The paper could benefit from studies to show how context benefits the model’s performance. The current approach of sampling 5% of the training data does not give much intuition of how the proposed helps improve the performance.
Proposed approach relies on a specific model architecture, limiting its applications to other model architectures.


**Summary Of The Paper:**

The paper proposed a novel few-shot learning approach (Hopfield-based molecular context enrichment for few-shot drug discovery or MHNfs) that exploits a context of other compounds outside the support and query set to improve the performance of the model. The model relies on the proposed context-module, made up of Hopfield layers, to associate the representations of the query and support set samples with other samples in the training data to enrich the representations. Afterwards, a cross-attention module is used to associate the representations within the query and support set. Experiments are conducted on FS-MOL and Tox21 datasets to show the proposed MHNfs is competitive versus state-of-the-art baselines and ablations experiments are conducted to show how the two modules improve the model performance.

**Summary Of The Review:**

The paper proposed several novel contributions to the molecular few-shot learning literature, especially its context retrieval module. Though there are still some questions about the proposed context retrieval module such as its applications to other model architecture, scalability and intuition behind its benefit, the experimental results are convincing.

---

> ### Author Response · Authors · 2022-11-15
> **Answer to reviewer b2MR**
>
> We thank the reviewer for all the work and valuable feedback which helps to enhance our manuscript. We are glad the reviewer assessed our approach to be novel and our experimental results to be convincing.
>
> ### Context module and context set size
> We thank the reviewer for these thoughts. Indeed, we hypothesize that it might be possible to improve the 5 % sampling strategy in future. The basic idea is to associate the input molecules  with a larger set of reference molecules. Intuitively one would think about using the whole training set as the external memory. Only because we had run into memory limitations, we performed the 5% sampling strategy. Since we focused on using a context set which is as large as possible, we did not perform any experiments with smaller context set sizes.
> Section A8 (p. 28-29) shows how the MHN reinforces the covariance structure and therefore explains why the context module helps. The ablation study  (Section 5.2, p. 8-9; Section A3.3, p. 25) shows that it significantly helps to improve the model performance.
> The context module, as well as the whole model architecture can directly be applied to any other few-shot scenario as long as there are embeddings available for the query, support set, and context instances.
> Lastly, we want to refer to concurrent work that uses a conceptually similar approach ( “Relative representations enable zero-shot latent space communication”, https://openreview.net/forum?id=SrC-nwieGJ). In this work, the authors use a set of instances, so-called anchors, to represent data in the latent space based on these anchors, which would be roughly analogous to our context molecules. Although the authors pursued a different research question, the set of anchors could be seen as an external memory. The authors mention that the question of the best choice of the anchor set still needs further exploration — just as we do regarding the context set. Nevertheless, the authors include a figure (Figure 6 in https://openreview.net/forum?id=SrC-nwieGJ) which shows the relation between model performance and the size of the anchor set. We will investigate this dependency more and add a Figure in the final version of the manuscript.
>
> ### Tox21 experiment
> The choice of the context set remains independent from the specific task for which predictions are desired. Therefore, for the Tox21 domain shift experiment, we used the context set like we used it for the FS-Mol benchmarking experiment, i.e. sampling 5 % from the FS-Mol training data. We also point to Section A.5., in which we explore different choices of the context set.
>
> ### Computational costs
> Comparing the training of MHNfs and “MHNfs -CM”, which is the MHNfs architecture without the context module, the context module comes with additional computational costs, which are a) additional costs needed to create the embeddings for the context molecules and b) the time needed for the Modern Hopfield related operations. a) can be precomputed and stored, which is why we just have to consider b) for inference time. So, the computational costs for predicting the label of one query molecule with support set size 16 are 0.0918 sec, whereas MHNfs-CM just needs 0.0037 sec (factor ~25). However, inference time is usually of lower relevance for applications in drug discovery, since the wet-lab experiments to determine the labels of the support set (i.e. active or inactive) require several days or weeks and, thus, are on a completely different time scale. A computational model would typically be used to virtually screen a library of physically available molecules, e.g. of size in the range of 100K molecules, for potential active molecules. This would take around 2.5 hours (1e6*0.09/3600) with MHNfs, which would be both acceptable and at the same time easily reducible, e.g. by parallelization.
>
> ### Applicability to other domains
> Indeed, we believe that the context module could be applied in other application domains and architectures. For computer vision tasks, a context module with tens of thousands of instances might still be computationally infeasible, but might already work for industrial manufacturing, logistics, and some areas of reinforcement learning. We also elaborate on this in Section A.9 (“Broader Impact”, p. 30).

---

### Author Response · Authors · 2022-11-15
**General response**

We thank the reviewers for their thorough and thoughtful reviews, which contained highly constructive, positive feedback that helped us to improve the manuscript. The reviewers acknowledged our work as novel in several aspects. They assessed our work to be of high quality standards and the presentation to be clear and well written. The results were considered convincing and the newly added baselines were appreciated.

The main point of concern was the overlapping error-bars between the methods. Indeed, the error bars of the FS-Mol benchmark are counter-intuitive, because one method can clearly outperform another while the error bars are still overlapping. We apologize that we were not able to explain this clearer. We tried to explain the error bars in the caption of Table 1, but we will now improve this description in the new version of the manuscript. We thank the reviewers for raising this point because it gave us the opportunity to add clarity to the paper in this regard and, therefore, to improve the manuscript.

In fact, the reported error bars on the metrics comprise *three sources of variation*:
- Variability across different 157 tasks. This is the largest source of variation.
- Variability across different support sets. During inference time, for each task, there are multiple draws of the support sets on which the models are evaluated.
- Variability across different training reruns. We trained the models multiple times and include this variability in the results.

In the FS-Mol benchmarking setting, we follow Stanley et al. which is why the error-bars report variability across tasks. Because of the variation across tasks the reported error bars in this table overlap. Nevertheless, the performed Wilcoxon test clearly indicates that one method clearly outperforms the other. The error bars for the experiments are analogous.

For further clarifying the sources of variation of the metrics and the method comparison:
- We added a table (Section A.2, p. 24) in which variability across training reruns is reported. In this table the overlap of the error bars is marginal.
- We added two scatterplots (Section A.2, p. 24) comparing the first method with its runner-up methods. These scatterplots show the variability across tasks (location of the points along x- and y-axis).

We answer the other comments and concerns of the reviewers individually below.
We uploaded a new version of the manuscript with the edits and new additions colored in orange.

---

### Decision · Program_Chairs · 2023-01-20

**Decision:**

Accept: poster

**Justification For Why Not Higher Score:**


The approach is simple yet effective. Technical novelty of any particular component is not high but they are put together in a clean way.

**Justification For Why Not Lower Score:**


The paper is very clearly written. The motivations for the various components are well-articulated (though largely intuitively) and well-explored through experiments. A number of baselines are included in the comparisons. The method gives state of the art performance on the FS-Mol benchmark.

**Metareview: Summary, Strengths And Weaknesses:**

The authors propose to solve a few-shot molecular property prediction problem by using a new context module to enrich molecular representations for the query and the support set. These contextual representations are derived from a simple associative memory implemented by a (multi-head) modern Hopfield network that takes in a larger reference set of molecules. The predicted answer to the query is then derived from cross-attention and similarity modules operating on the new query/support set representations. The approach is simple yet effective, the motivations for the various components are well-articulated (largely intuitively) and tested through ablation experiments. A number of baselines are included. The method gives state of the art performance on the FS-Mol benchmark.


**Note From Pc:**

if the above contains the word "oral" or "spotlight" please see: "oral" presentation means -> notable-top-5% and "spotlight" means -> notable-top-25%. As stated in our emails, we are disassociating presentation type from AC recommendations